# The Gaussian Mixing Mechanism: Rényi Differential Privacy via Gaussian Sketches

**Omri Lev**[1*] **Vishwak Srinivasan**[1,†] **Moshe Shenfeld**[2,†]

**Katrina Ligett**[2] **Ayush Sekhari**[3] **Ashia C. Wilson**[1]

[1] Massachusetts Institute of Technology
[2] The Hebrew University of Jerusalem    [3] Boston University

## Abstract

Gaussian sketching, which consists of pre-multiplying the data with a random Gaussian matrix, is a widely used technique in data science and machine learning. Beyond computational benefits, this operation also provides differential privacy guarantees due to its inherent randomness. In this work, we revisit this operation through the lens of Rényi Differential Privacy (RDP), providing a refined privacy analysis that yields significantly tighter bounds than prior results. We then demonstrate how this improved analysis leads to performance improvement in different linear regression settings, establishing theoretical utility guarantees. Empirically, our methods improve performance across multiple datasets and, in several cases, reduce runtime.

## 1 Introduction

Advances in data collection and storage have enabled the creation of massive datasets, which have necessitated the design of scalable algorithms for efficient processing, analysis, and inference. With the formation of such large datasets, there has also been an increased focus on privacy-preserving analyses in a bid to protect private attributes [Apple Research, 2017, Facebook Research, 2020, Snap Security, 2022, Ponomareva et al., 2023]. For instance, census data consists of a variety of private information about individuals recorded, and the United States Census Bureau has been taking various measures to ensure confidentiality in the data, and has adopted more modern methods since the late 1900s [Bureau, 2019].

A popular mathematical notion of privacy is given by *Differential Privacy* (DP) [Dwork, 2006, McSherry and Talwar, 2007, Dwork et al., 2014], which is currently the de facto standard for privacy-preserving mechanisms. A private mechanism is a (randomized) function $\mathcal{M}$ which takes as input a dataset $X$, and the key challenge in designing a "good" private mechanism is balancing the tradeoff between the loss in utility with the increase in privacy-preserving nature of the algorithm; it is possible to obtain complete privacy by simply returning noise regardless of the input, but this clearly does not have good utility.

With regard to designing scalable algorithms to handle massive datasets, *sketching* as a technique [Sarlos, 2006] has led to more computationally scalable algorithms relative to naïvely working with the full dataset; see Woodruff [2014] for a review. Given a data matrix $X \in \mathbb{R}^{n \times d}$, a *sketch* constructs a compressed representation $\mathsf{S}X \in \mathbb{R}^{k \times d}$ with $k \ll n$, where $\mathsf{S} \in \mathbb{R}^{k \times n}$ is a random matrix. For the case where $\mathsf{S}$ is comprised of i.i.d. Gaussian elements,

---

*Correspondence: `omrilev@mit.edu`. Code: `https://github.com/omrilev1/GaussMix`.
†Equal contribution.

for sufficiently large $k$ it holds that $\frac{1}{k}\mathsf{S}^\top\mathsf{S} \approx \mathbb{I}_n$ with high probability. This property is particularly valuable in machine learning applications that rely on inner-products of the form $A^\top B$ for matrices $A$ and $B$, as one could now apply a Gaussian sketching matrix $\mathsf{S}$ to both $A$ and $B$ and largely preserve this Hilbert-Schmidt inner product (up to scaling constants). In this work, we study a mechanism based on a "noisy" sketching operation, which we call the *Gaussian Mixing Mechanism* (abbrev. GaussMix) and is defined by

$$\mathcal{M}(X) := \mathsf{S}X + \sigma\xi\ , \quad \text{with } \mathsf{S} \sim \mathcal{N}(0, \mathbb{I}_{k \times n}),\ \xi \sim \mathcal{N}(0, \mathbb{I}_{k \times d}),\ S \perp \xi \qquad \text{(GaussMix)}$$

The additive Gaussian noise (hence the "noisy" sketching operation) essentially contributes a constant bias to the inner product between the outputs of GaussMix applied to $A$ and $B$, as for sufficiently large $k$ and appealing to standard concentration inequalities, we have

$$\frac{1}{k}(\mathsf{S}A + \sigma\xi)^\top(\mathsf{S}B + \sigma\xi) \approx A^\top B + \sigma^2\mathbb{I}_d\ . \tag{1}$$

Thus, this noisy sketch is potentially well-suited for applications involving inner products. The intriguing aspect of GaussMix is that the Gaussian sketching operation contributes to stronger privacy guarantees when compared to the standard Gaussian mechanism.

The first evidence of sketching yielding a differentially private mechanism was given by Blocki et al. [2012], and later works [Sheffet, 2017, 2019] focused on establishing that GaussMix also provides DP. While it might seem surprising that GaussMix is a differentially private mechanism, the operation can be viewed as instantiating the classical *Gaussian mechanism* for DP [Dwork et al., 2014, Appendix A] for the Gaussian sketched data matrix $\mathsf{S}X$.

**Intuition for GaussMix.** Interestingly, certain key operations in augmenting mechanisms for improved differential privacy can be expressed as left-multiplication operations. Two such instances are (1) permuting the rows of the data matrix $X$ randomly [Erlingsson et al., 2019], which can be expressed as left-multiplication of $X$ by a permutation matrix, and (2) subsampling the data matrix at random, which can be expressed as left-multiplication of $X$ by a random matrix whose entries are either 0 or 1 [Kasiviswanathan et al., 2011, Balle et al., 2018]. A rough intuition for why sketching is a reasonable operation for preserving privacy is that $\mathsf{S}X$ generates random linear combinations of rows, which can potentially hide the contribution of any one row. Differential privacy—a more rigorous notion—seeks to ensure that the presence of any one particular row in the data can not be guessed well, even by an adversary that has knowledge of the other rows. Note that sketching by itself is not impervious to such a setting: consider a data matrix $X$ where all but one row contain 0s, and in this case, it is possible to make educated guesses for what the non-zero row might be from the sketched data matrix $\mathsf{S}X$. This motivates the inclusion of the noise addition in GaussMix, which also covers this adversarial situation. Informally speaking, through our analysis, we observe that the "richness" of the data matrix (quantified by the minimum singular value) contributes to the success of the sketching mechanism as a differentially private mechanism.

**Contributions.** Expanding on the above, our primary contributions, which are both theoretical and empirical, are summarized below.

- **Tighter Privacy Analysis via Rényi Differential Privacy (RDP).** We present a new RDP analysis of GaussMix, which to the best of our knowledge, has not been previously explored. Our bounds are simpler to derive than existing analyses and in several cases, are tight with respect to RDP. Notably, our results show that GaussMix achieves stronger privacy guarantees than those established by Sheffet [2019] for the same mechanism parameters. Our imprvement extend to settings that rely on GaussMix, such as Prakash et al. [2020], Anand et al. [2021], Sun et al. [2022], Bartan and Pilanci [2023].

- **Algorithm for Private Ordinary Least Squares.** We use GaussMix to develop a differentially private algorithm for linear regression under bounded data, extending the framework of Sheffet [2017]. Leveraging our new RDP analysis, we derive bounds on the excess empirical risk that, in some cases, match those of the AdaSSP algorithm by Wang [2018], a standard benchmark under domain-bound assumptions. We validate these theoretical results through empirical evaluations, where our method consistently outperforms the baselines of Sheffet [2017] and Wang [2018] across several benchmark

datasets. Furthermore, we show that retrofitting the improved RDP analysis into the algorithm of Sheffet [2017] yields improved results, further demonstrating the broad applicability of our improved analysis.

- **Algorithm for Private Logistic Regression.** We adapt the algorithm for private linear regression to perform differentially private logistic regression. This specifically works by using a second-order approximation of the loss function [Huggins et al., 2017, Ferrando and Sheldon, 2025] that reduces the problem to a differentially private quadratic minimization, which makes it amenable to apply GaussMix. We derive theoretical guarantees for our proposed method and give numerical simulations over certain datasets that demonstrate improvements in both accuracy and computation time over the commonly used objective perturbation [Chaudhuri et al., 2011] and DP-SGD [Abadi et al., 2016] baselines.

**Organization of the paper.** In Section 2, we discuss related work. Section 3 introduces necessary preliminaries. We present the Gaussian Mixing Mechanism and our privacy analysis in Section 4, including comparisons with prior bounds. Applications of our mechanism to DP OLS (Section 5.1) and DP logistic regression (Section 5.2) are given in Section 5, supported by theoretical and empirical evaluations.

## 2   Related Work

A substantial body of research has investigated the use of random matrix projections for privacy, particularly through the Johnson–Lindenstrauss (JL) transform and its variants [Blocki et al., 2012, Kenthapadi et al., 2013, Sheffet, 2017, Showkatbakhsh et al., 2018, Sheffet, 2019]. Sheffet [2015, 2017, 2019] and Showkatbakhsh et al. [2018] are the most relevant to our work, as they propose using Gaussian sketches for private linear regression. Sheffet [2015, 2017, 2019] analyze the privacy-preserving characteristics of GaussMix for this problem, and show that it achieves $(\varepsilon, \delta)$-differential privacy for certain settings of $\sigma, k$. Showkatbakhsh et al. [2018] study the same mechanism but under a modified notion of differential privacy known as MI-DP (defined by Cuff and Yu [2016]).

The mathematical intuition for why sketching is useful in privacy-sensitive optimization was studied by Pilanci and Wainwright [2015]. They observe that the mutual information between $X$ and its sketched version $SX$ can not be too large, thus providing a form of privacy. However, their analysis is centered around information-theoretic principles and quantities such as the mutual information, and does not actually provide guarantees of differential privacy. More recently, Bartan and Pilanci [2023] provides another application of Gaussian sketching in the specific context of DP distributed linear regression, based on the results obtained by Sheffet [2015].

In contrast to these prior approaches, our work uses the stronger RDP framework of Mironov [2017] in its privacy analysis. We derive a simple, closed-form expression for the RDP curve $\varepsilon(\alpha)$ corresponding to GaussMix, which is tight and improves upon the bounds obtained in earlier works. These improvements yield practical gains in the performance of our algorithm for DP linear regression relative to both Sheffet [2019] and other common alternatives such as Wang [2018], and can be further used in other settings that currently use (GaussMix), such as Prakash et al. [2020], Anand et al. [2021], Sun et al. [2022], Bartan and Pilanci [2023]. Moreover, we further derive a computationally efficient algorithm for private logistic regression. Our numerical experiments demonstrate that these enhancements translate into improved accuracy over standard baselines.

## 3   Preliminaries

**Basic Notation.** We denote random variables in sans-serif (e.g., $\mathsf{X}, \mathsf{y}$), and their realizations in serif (e.g., $X, y$ resp.). The set $\{1, \ldots, n\}$ is denoted by $[n]$. For a vector $A \in \mathbb{R}^d$ its Euclidean norm is denoted by $\|A\|$. The all-zeros column vector of length $d$ is denoted by $\vec{0}_d$. The $k \times k$ identity matrix is $\mathbb{I}_k$ and $\mathcal{N}(0, \mathbb{I}_{k_1 \times k_2})$ denotes a $k_1 \times k_2$ matrix of i.i.d. standard Gaussian entries. We give a more elaborate discussion of notation in Appendix A.

**Differential Privacy.** Differential privacy relies on the notion of a "neighboring" dataset, which we introduce first. Two datasets $X, X'$ are called *neighbors* if $X'$ is formed by removing an element from $X$ [2] or vice-versa, and we use $X \simeq X'$ to denote this relation. In this work, a dataset is regarded as a collection of $n$ real-valued rows, each of length $d$ for $n, d \geq 1$. For a dataset given to us, we assume knowledge of an upper bound $C_X$ (called the *row bound*) where $\|x_i\| \leq C_X$ for all $i \in [n]$.

Intuitively, differential privacy, formalized in the next definition, requires that a randomized algorithm induce nearly identical output distributions given neighboring input datasets.

**Definition 1** $((\varepsilon, \delta)$-Differential Privacy [Dwork et al., 2006]). A randomized mechanism $\mathcal{M}$ is said to satisfy $(\varepsilon, \delta)$-*differential privacy* if for all $X, X'$ such that $X' \simeq X$ and measurable subsets $\mathcal{S} \subseteq \text{Range}(\mathcal{M})$,

$$\Pr(\mathcal{M}(X) \in \mathcal{S}) \leq e^\varepsilon \cdot \Pr(\mathcal{M}(X') \in \mathcal{S}) + \delta .$$

A secondary, somewhat stronger notion of differential privacy that we adopt throughout this work is given by Rényi-DP, first introduced by Mironov [2017].

**Definition 2** $((\alpha, \varepsilon(\alpha))$-RDP [Mironov, 2017]). A randomized mechanism $\mathcal{M}$ is said to satisfy $(\alpha, \varepsilon(\alpha))$-RDP for some $\alpha > 1$ if for all $X, X'$ such that $X \simeq X'$,

$$D_\alpha\left(\mathcal{M}(X) \,\|\, \mathcal{M}(X')\right) \leq \varepsilon(\alpha) ,$$

where $D_\alpha(P \,\|\, Q) := \frac{1}{\alpha-1} \log\left(\mathbb{E}_{x \sim Q}\left[\left(\frac{P(x)}{Q(x)}\right)^\alpha\right]\right)$ denotes the $\alpha$-Rényi divergence [Rényi, 1961].

The notion of $(\varepsilon, 0)$-DP can be viewed as ensuring that the likelihood ratio of events induced by neighboring datasets are uniformly bounded, and $\delta$ in $(\varepsilon, \delta)$-DP provides some additive slack on this condition. On the other hand, for any $\alpha > 1$, $(\alpha, \varepsilon(\alpha))$-Rényi-DP can be seen as another control that bounds the moments of this likelihood ratio. The latter can be translated into the former; this conversion is explicitly stated in the proposition below.

**Proposition 1** (Canonne et al. [2020, Proposition 12]). *If $\mathcal{M}$ satisfies $(\alpha, \varepsilon(\alpha))$-RDP , then it also satisfies $(\varepsilon_{\text{DP}}, \delta)$-DP for any $0 < \delta < 1$, where $\varepsilon_{\text{DP}} = \varepsilon(\alpha) + \log\left(1 - \frac{1}{\alpha}\right) - \frac{\log(\alpha\delta)}{(\alpha-1)}$.*

Both $(\alpha, \varepsilon(\alpha))$-RDP and $(\varepsilon, \delta)$-DP satisfy key properties such as graceful degradation under composition and post-processing. In particular, the post-processing property ensures that if a mechanism $f$ satisfies either privacy definition, then so does $g \circ f$ for any (possibly randomized) function $g$ [Dwork et al., 2014, Mironov, 2017].

We highlight a special family of mechanisms that satisfy $(\alpha, \varepsilon(\alpha))$-RDP for a range of values of $\alpha$ wherein $\varepsilon(\alpha)$ grows at most linearly in $\alpha$ within this range. Such mechanisms are said to satisfy *truncated concentrated DP* (tCDP) which is defined formally below.

**Definition 3** (tCDP [Bun et al., 2018]). Let $\rho > 0$ and $w > 1$. A mechanism $\mathcal{M}$ satisfies $(\rho, w)$-tCDP if $D_\alpha(\mathcal{M}(X)\|\mathcal{M}(X')) \leq \rho \cdot \alpha$ for all neighboring $X, X'$ and for all $\alpha \in (1, w)$.

The $(\rho, w)$-tCDP property lends to a tighter translation to $(\varepsilon_{\text{DP}}, \delta)$-DP in comparison to simply picking $\alpha \in (1, w)$ and instantiating Proposition 1. This fact leads to tighter privacy guarantees for GaussMix as we present in the latter parts of this paper.

**Proposition 2** (Bun et al. [2018, Lemma 6]). *If $\mathcal{M}$ satisfies $(\rho, w)$-tCDP, then its also satisfies $(\varepsilon_{\text{DP}}, \delta)$-DP for all $\delta > 0$ where*

$$\varepsilon_{\text{DP}} = \begin{cases} \rho + 2\sqrt{\rho \cdot \log(1/\delta)} & \text{if } \log(1/\delta) \leq (w-1)^2 \cdot \rho \\ \rho \cdot w + \frac{\log(1/\delta)}{w-1} & \text{if } \log(1/\delta) > (w-1)^2 \cdot \rho . \end{cases}$$

**Gaussian mechanism.** The *Gaussian mechanism* is a standard baseline for achieving $(\varepsilon, \delta)$-DP by simply adding Gaussian noise to (some function of) the data before releasing it. In our notation it amounts to $\mathcal{M}_G(X) = X + \sigma\xi$ with $\xi \sim \mathcal{N}(0, \mathbb{I}_{n \times d})$. The Gaussian mechanism is $\left(\frac{C_X^2}{2\sigma^2}, \infty\right)$-tCDP (also known as zCDP [Bun and Steinke, 2016]), and $(\varepsilon, \delta)$-DP where $\varepsilon = \frac{\sqrt{2\log(1.25/\delta)}}{\sigma}$ for any $\delta \in (0, 1)$ [Dwork et al., 2014, Appendix A].

---

[2] For simplicity, we identify a removal of a row with its replacement by $\vec{0}_d$, so the dimension remains constant. This notion is sometimes referred to as *zero-out* neighboring.

## 4 The Gaussian Mixing Mechanism

We start by providing a new privacy analysis of GaussMix, under the assumption that we have a lower bound $\overline{\lambda}_{\min}$ for the minimum eigenvalue of $X^\top X$ (called the *scale bound*).

**Lemma 1.** *For any data matrix $X \in \mathbb{R}^{n \times d}$ that satisfies row bound $C_X$ and scale bound $\overline{\lambda}_{\min}$, GaussMix with parameters $k$ and $\sigma$ such that $\gamma := C_X^{-2} \cdot (\sigma^2 + \overline{\lambda}_{\min}) > 1$ satisfies $(\alpha, \varphi(\alpha; k, \gamma))$-RDP for any $\alpha \in (1, \gamma)$, where $\varphi(\alpha; k, \zeta) := \frac{k\alpha}{2(\alpha-1)} \log\left(1 - \frac{1}{\zeta}\right) - \frac{k}{2(\alpha-1)} \log\left(1 - \frac{\alpha}{\zeta}\right)$.*

To help understand the role of $\gamma$, this can be viewed as a lower bound on the minimal eigenvalue of the matrix $\widetilde{X}^\top \widetilde{X}$ where $\widetilde{X} := C_X^{-1} \cdot [X^\top, \sigma \mathbb{I}_d]^\top \in \mathbb{R}^{(n+d) \times d}$. Using this perspective, the noise addition can be seen as a way to artificially raise the minimum singular value of the matrix $X$ to a predetermined threshold $\gamma$, after which a standard Gaussian sketching step is applied. This reinforces the prior intuition that privacy arises from applying Gaussian sketching to a matrix with a sufficiently large minimum singular value.

*Proof sketch of Lemma 1.* For a fixed $X$, we first deduce that every row of $\mathcal{M}(X)$ is distributed according to a multivariate Gaussian distribution with zero mean and covariance $\Sigma := X^\top X + \sigma^2 \mathbb{I}_d$. Using the closed form expression of the Rényi divergence between multivariate Gaussian distributions, the quantity $D_\alpha(\mathcal{M}(X) \| \mathcal{M}(X'))$ is a monotonic function of $x^\top \Sigma^{-1} x$ where $x$ is the row in $X$ that is zeroed out in $X'$. This quantity can be bounded as $x^\top \Sigma^{-1} x \leq (\lambda_{\min}(\Sigma))^{-1} \cdot \|x\|^2$, and $\lambda_{\min}(\Sigma) \geq \overline{\lambda}_{\min} + \sigma^2$ by the scale bound. We defer the details of this proof to Appendix C.1.

Two key observations from the analysis is that (1) the function $\varphi$ is non-negative for every $\alpha \in (1, \gamma)$ and further bounds the RDP curve of $\mathcal{M}(X)$ from above, and (2) for sufficiently large $\gamma$, the function $\varphi$ in Lemma 1 grows at most linearly in $\alpha$ in a certain range. The latter property is precisely the definition of tCDP (Definition 3).

**Corollary 1.** *Consider the setup of Lemma 1. If $\gamma > 5/2$, then GaussMix satisfies $(k/2\gamma^2, 2\gamma/5)$-tCDP.*

**Tightness of Bound.** In our computations, the only inequality that occurs is the one stated in the proof sketch. This inequality is tight when the row in which $X$ differs from $X'$ happens to be the eigenvector corresponding to the minimal eigenvalue of $X^\top X$. This implies that the lemma is tight for specific inputs, ensuring the bound is optimal under certain input assumptions. More precisely, for the set of matrices $X \in \mathbb{R}^{n \times d}$ such that $n \geq d$ with norm bounds $C_X$ and such that $\lambda_{\min}(X^\top X) \geq C_X^2$, our bound is achieved with equality for any $X$ such that $X^\top X = C_X^2 \cdot \mathbb{I}_d$, corresponding to the case where $\frac{1}{C_X} X$ is a semi-orthogonal matrix.

**Comparison to Existing Literature.** Below we summarise the comparison between our guarantees for the $(\varepsilon, \delta)$-DP of GaussMix and a prior result by Sheffet [2019].

|  | Our result | Sheffet [2019] |
|---|---|---|
|  | (Corollary 1 + Proposition 2) | (Theorem 2) |
| $\varepsilon$ | $\frac{k}{2\gamma^2} + \frac{\sqrt{2k \log(1/\delta)}}{\gamma}$ | $\frac{2\sqrt{2k \log(4/\delta)}}{\gamma} + \frac{2 \log(4/\delta)}{\gamma}$ |

For a fixed $k$, the dependence on $\gamma$ in our guarantee is strictly better than the result of Sheffet [2019]; specifically the bound from Sheffet [2019] suffers from the additional $\frac{2 \log(4/\delta)}{\gamma}$ term. Numerical analysis based on the exact RDP guarantee (derived by using the conversion from RDP to DP of Proposition 1) provides a larger improvement of Lemma 1 over the bound of Sheffet [2019]; see Figure 1 for plots to showcasing this comparison.

Moreover, we note that with GaussMix the entire dataset contributes to the privacy protection of a single element via mixing. Since $\overline{\lambda}_{\min}$ is at most $\frac{nC_X^2}{d}$ (see discussion in [Wang, 2018]), this resembles privacy amplification by shuffling or subsampling, whereby the added noise to other elements contributes to the privacy protection of any element. The dependence of $\varepsilon$ on the parameters is not trivially comparable since $k$ affects only GaussMix, and the utility of the two outputs might be arbitrarily different.

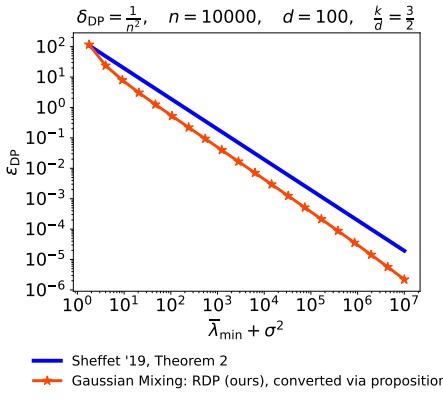
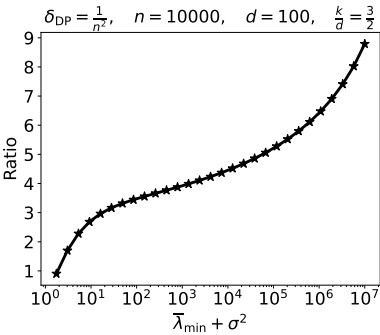

(a) Required $\varepsilon_{\mathrm{DP}}$.

(b) Ratio between the bounds.

Figure 1: $\overline{\lambda}_{\min} + \sigma^2$ required for attaining $\varepsilon_{\mathrm{DP}}$, using Lemma 1 converted to $(\varepsilon_{\mathrm{DP}}, \delta)$-DP via Proposition 1, and compared to the bound from Sheffet [2019, Theorem 2]. We plot (a) $\varepsilon_{\mathrm{DP}}$ as a function of $\overline{\lambda}_{\min} + \sigma^2$ and (b) the ratio between the $\varepsilon_{\mathrm{DP}}$ bounds. Our improved RDP analysis reduces the final $\varepsilon_{\mathrm{DP}}$, illustrating the benefit of our new approach.

---

**Algorithm 1** ModifiedGaussMix

---

**Require:** Dataset $X \in \mathbb{R}^{n \times d}$, row bound $\mathrm{C}_X$, parameters $k, \gamma, \tau, \eta$.
1: **if** $\gamma \leq \tau$ **then**:
2:     **Output:** $\mathsf{S}X + \gamma \mathrm{C}_X \xi_1$ with $\mathsf{S} \sim \mathcal{N}(0, \mathbb{I}_{k \times n})$ and $\xi_1 \sim \mathcal{N}(0, \mathbb{I}_{k \times d})$.
3: **else**:
4:     Set $\widetilde{\lambda} = \max\left\{\lambda_{\min}(X^\top X) - \eta \mathrm{C}_X^2 (\tau - \mathsf{z}), 0\right\}$, where $\mathsf{z} \sim \mathcal{N}(0, 1)$.
5:     Set $\widetilde{\eta} = \sqrt{\max\{\gamma - \widetilde{\lambda}, 0\}}$.
6:     **Output:** $\mathsf{S}X + \widetilde{\eta} \mathrm{C}_X \xi_1$ with $\mathsf{S} \sim \mathcal{N}(0, \mathbb{I}_{k \times n})$ and $\xi_1 \sim \mathcal{N}(0, \mathbb{I}_{k \times d})$.

---

**Usage Without $\overline{\lambda}_{\min}$.** Since the bounded scale assumption cannot be enforced in the general case, one way to utilize Lemma 1 is by simply using the fact that $\overline{\lambda}_{\min} \geq 0$ and relying solely on the added noise. In this case, the GaussMix mechanism can be thought of as a complicated counterpart of the Gaussian mechanism. Comparing their privacy guarantees reveals a clear similarity, with $\sqrt{k}/\gamma$ taking the role of $1/\sigma$ in the Gaussian mechanism. In this case $\gamma = \sigma^2$ which amounts to a quadratic dependence of $\varepsilon$ on $\sigma$ in the leading term (rather than the linear one in the Gaussian mechanism), balanced by $\sqrt{k}$, which implies improved privacy results when $\sigma > \sqrt{k}$. Since the output of these two mechanisms follows different distributions, the privacy-utility tradeoff depends on the use case.

**Instance Specific Bound.** While setting $\overline{\lambda}_{\min} = 0$ provides a privacy guarantee for any matrix, one might wish to reduce the added noise in a data-dependent manner based on $\lambda_{\min}(X^\top X)$. Since this quantity is data-dependent, it must be used in a privacy-preserving manner, which is achieved using Algorithm 1. A similar algorithm was proposed by Sheffet [2017, Algorithm 1], with $\mathsf{z}$ sampled from a Laplace distribution, and $\eta$ set either to 0 or to $\sqrt{\gamma}$ based on $\widetilde{\lambda}$, rather than interpolating between the two as we do.

**Theorem 1.** *Let $\delta \in (0, 1)$, $k \geq 1$, $\eta > 0$ and $\gamma > {}^5\!/_2$, and set $\tau \geq \sqrt{2 \log({}^3\!/_\delta)}$. Then, the output of Algorithm 1 is $(\widetilde{\varepsilon}(\eta, \gamma, k, \delta), \delta)$-DP for*

$$\widetilde{\varepsilon}(\eta, \gamma, k, \delta) = \frac{\sqrt{2 \log({}^{3.75}\!/_\delta)}}{\eta} + \min_{1 < \alpha < \gamma} \left\{ \varphi(\alpha; k, \gamma) + \frac{\log({}^3\!/_\delta) + (\alpha - 1) \log(1 - {}^1\!/_\alpha) - \log(\alpha)}{\alpha - 1} \right\} \tag{2}$$

*where $\varphi(\alpha; k, \gamma)$ was previously defined in Lemma 1.*

*Proof sketch.* The case of $\gamma < \tau$ is the previously discussed $\overline{\lambda}_{\min} = 0$ setting. For the other case, we note that w.p. $\geq 1 - \delta/3$ we have $\widetilde{\lambda} \geq \lambda_{\min}(X^\top X)$, so the conditions of Lemma 1 are met. Since $\overline{\lambda}_{\min}$ has sensitivity $C_X^2$, the privacy properties of the Gaussian mechanism apply to the release of $\widetilde{\lambda}$. Using simple composition, we combine it with the privacy of GaussMix, which completes the proof. We give full details in Appendix C.3.

The quantity $\widetilde{\varepsilon}(\eta, \gamma, k, \delta) \geq 0$ and further can be made arbitrarily small by increasing $\eta$ and $\gamma$; we show this formally in Appendix B. As mentioned previously, the parameter $\gamma$ should be thought of as the target minimal eigenvalue, $\tau$ serves as the accuracy bound on the estimation of $\overline{\lambda}_{\min}$, and $\eta$ controls the privacy loss of its private estimation. Setting $\eta = \gamma/\sqrt{k}$, and combining Corollary 1 with the DP implication of tCDP, we get that Algorithm 1 is $(\varepsilon, \delta)$-DP where $\varepsilon = \frac{k}{2\gamma^2} + \frac{2\sqrt{2k\log(4/\delta)}}{\gamma}$, which matches the privacy bound on GaussMix up to a constant.

## 5 Applications

We now demonstrate applications of GaussMix for two private learning tasks that can be formalized as linear regression. In each case, we instantiate the mechanism on a concatenated dataset $(X_1, X_2)$ where $X_1 \in \mathbb{R}^{n \times d_1}$ and $X_2 \in \mathbb{R}^{n \times d_2}$. The application of GaussMix to the concatenation $(X_1, X_2)$ is given by

$$\mathcal{M}(X_1, X_2) = \mathsf{S}(X_1, X_2) + \sigma(\xi_1, \xi_2) = (\mathsf{S}X_1 + \sigma\xi_1, \mathsf{S}X_2 + \sigma\xi_2)$$

where $\mathsf{S} \sim \mathcal{N}(0, \mathbb{I}_{k \times n})$, $\xi_1 \sim \mathcal{N}(0, \mathbb{I}_{k \times d_1})$, and $\xi_2 \sim \mathcal{N}(0, \mathbb{I}_{k \times d_2})$. If $\mathcal{M}(X_1, X_2)$ satisfies RDP w.r.t. $(X_1, X_2)$, then by post-processing the inner-product $\mathcal{K}$ defined as

$$\mathcal{K}(\mathcal{M}(X_1, X_2)) := (\mathsf{S}X_1 + \sigma\xi_1)^\top (\mathsf{S}X_2 + \sigma\xi_2) \tag{3}$$

also satisfies RDP . The inner-product $\mathcal{K}$ will form the core component of our algorithms.

### 5.1 Differentially Private Ordinary Least Squares

We begin with the problem of DP linear regression. Let the dataset $(X, Y)$ such that the design matrix $X \in \mathbb{R}^{n \times d}$ and the response vector $Y \in \mathbb{R}^n$. Throughout, we make the following assumptions:

($\mathbf{A}_1$) *Bounded domain:* $\|x_i\| \leq C_X$ and $|y_i| \leq C_Y$ for all $i \in [n]$[3].
($\mathbf{A}_2$) *Overspecified system:* $n \geq d$.

Our goal is to estimate a linear predictor under $(\varepsilon, \delta)$-DP, while preserving the privacy of each individual data pair $(x_i^\top, y_i)$. Our non-private baseline is the standard Ridge regression estimator [Tikhonov, 1963, Hoerl and Kennard, 1970]:

$$\theta^*(\nu) := \operatorname*{argmin}_{\theta} \left\{ \|Y - X\theta\|_2^2 + \nu\|\theta\|_2^2 \right\} = (X^\top X + \nu\mathbb{I}_d)^{-1} X^\top Y,$$

where $\nu \geq 0$ is a regularization parameter. The unregularized least-squares solution is denoted $\theta^* := \theta^*(0)$, and we define the empirical loss as $L_{X,Y}(\theta) := \|Y - X\theta\|_2^2$. Our algorithm, summarized in Algorithm 2, uses Algorithm 1 for obtaining a privatized version of the pair $(X, Y)$ (interpreted as halves of one joint matrix) by setting a large enough $\gamma$ and then solving the resulting least-squares problem. The existence of a $\gamma$ that satisfies the conditions in Line 1 in Algorithm 2 is ensured by the analysis presented in Appendix B. While the structure of our algorithm resembles earlier proposal by Sheffet [2017], our refined privacy analysis under Rényi-DP improves the overall privacy–utility trade-off. Moreover, our algorithm exploits $\lambda_{\min}$ in a modified way, ensuring that the variance of the additive noise component is always reduced by utilizing the private estimation of $\lambda_{\min}(X^\top X)$.

**Corollary 2.** *For any $k \geq 1$ the output $\theta_{\mathrm{Lin}}$ of Algorithm 2 is $(\varepsilon, \delta)$-DP .*

---

[3]These domain bounds appear in many standard DP linear regression settings, such as [Sheffet, 2017, Wang, 2018].

---
**Algorithm 2** LinearMixing
---
**Require:** Dataset $(X, Y)$ satisfying $(\mathbf{A}_1, \mathbf{A}_2)$, privacy parameters $(\varepsilon, \delta)$, parameter $k$.
  1: Find smallest $\gamma > {}^5/_2$ such that $\widetilde{\varepsilon}(\eta, \gamma, k, \delta) \leq \varepsilon$, while setting $\eta = \gamma/\sqrt{k}$.
  2: Calculate $\left[\widetilde{X}, \widetilde{Y}\right] = \texttt{ModifiedGaussMix}\left([X, Y], \sqrt{\mathrm{C}_X^2 + \mathrm{C}_Y^2}, k, \gamma, \sqrt{2 \log\left({}^3/\delta\right)}, \eta\right)$.
  3: **Output:** $\theta_{\mathrm{Lin}} \coloneqq \left(\widetilde{X}^\top \widetilde{X}\right)^{-1} \widetilde{X}^\top \widetilde{Y}$.
---

*Proof.* We first note that for any $k \geq 1$, we are guaranteed to find a $\gamma$ in Line 1 such that the target $\widetilde{\varepsilon}(\eta, \gamma, k, \delta) \leq \varepsilon$ where $\eta = \gamma/\sqrt{k}$ (see Appendix B). Since we apply ModifiedGaussMix (Line 2) with the appropriate domain bounds, the differential privacy guarantee follows from Theorem 1. Finally, since $\theta_{\mathrm{Lin}}$ is a function of $(\widetilde{X}, \widetilde{Y})$, which is $(\varepsilon, \delta)$-DP, we have by post-processing that $\theta_{\mathrm{Lin}}$ also satisfies $(\varepsilon, \delta)$-DP. $\qquad\square$

**Theorem 2.** *There exist universal constants $c_0, c_1, c_2$ such that for any $\chi \in (0, 1]$ satisfying $k\chi^2 \geq c_0 d$, the following holds with probability at least $1 - c_1 \cdot \exp\left\{-c_2 k\chi^2\right\}$:*

$$L_{X,Y}(\theta_{\mathrm{Lin}}) - (1 + \chi)^2 \cdot L_{X,Y}(\theta^*) \leq O\left((1 + \chi)^2 \cdot \frac{\sqrt{k \log(1/\delta)} \cdot (C_X^2 + C_Y^2)}{\varepsilon} \cdot \left(1 + \|\theta^*\|^2\right)\right).$$

The proof is deferred to Appendix C.4. Notably, in the regime where $\mathrm{C}_Y \approx \mathrm{C}_X \|\theta^*\|$ and $d \gg 1$, for a target failure probability $\varrho$, setting $\chi = \sqrt{k^{-1} \cdot \max\left\{c_0 d, (1/c_2) \log(c_1/\varrho)\right\}}$ and $k = (1 + \beta) \max\left\{c_0 d, 1/c_2 \log(c_1/\varrho)\right\}$ for some finite $\beta > 0$ yields excess empirical risk of $O\left(\varepsilon^{-1}\sqrt{\max\left\{d, \log(1/\varrho)\right\} \log(1/\delta)} C_X^2 \left(1 + \|\theta^*\|^2\right) + \chi L_{X,Y}(\theta^*)\right)$. In particular, we note that the first term avoids the multiplicative $\log(d^2/\varrho)$ factor that exists in the guarantees from Wang [2018]. Thus, whenever it holds that $\lambda_{\min}(X^\top X) \ll 1$ and furthermore $L_{X,Y}(\theta^*) \ll \varepsilon^{-1}\sqrt{\max\left\{d, \log(1/\varrho)\right\} \log(1/\delta)} C_X^2 \left(1 + \|\theta^*\|^2\right)$ we expect our method to outperform the AdaSSP baseline. Conversely, in high-residual regimes—where our bound incurs the term $\chi L_{X,Y}(\theta^\star)$—or when $\lambda_{\min}(X^\top X)$ is large—so that AdaSSP's guarantee enjoy from this minimal eigenvalue in the denominator of its guarantees, AdaSSP is likely to perform better.

To demonstrate the usefulness of Algorithm 2, we have simulated its performance on four different datasets: the Communities & Crime dataset [Redmond and Baveja, 2002], the Tecator dataset [Thodberg, 2015], a synthetic dataset comprised of Gaussian features transformed via an MLP and another dataset comprised of Gaussian features. Full experimental details are provided in Appendix F. We use algorithm. 1 from [Sheffet, 2017] as well as the AdaSSP algorithm of [Wang, 2018] as our private baselines. We further simulated a variant of algorithm. 1 from [Sheffet, 2017] that uses the analysis established in Lemma 1. The AdaSSP algorithm is considered the leading baseline for DP linear regression under bounded domain assumptions [Liu et al., 2022, Brown et al., 2024]. While some recent works, such as [Ferrando and Sheldon, 2025], have proposed practical enhancements over AdaSSP, this method involves significantly higher computational cost and lacks theoretical utility guarantees, as opposed to our method which runs essentially within the same time constants. As Figure 2 shows, our method outperforms the AdaSSP method and the method of [Sheffet, 2017], achieving lower or equal test MSE across all privacy levels on every dataset. Furthermore, incorporating the analysis of Lemma 1 into the algorithm of [Sheffet, 2017] improves its performance, hinting on the further usefulness of our new analysis technique.

## 5.2 Differentially Private Logistic Regression

Our second application considers the problem of DP logistic regression, with the goal of developing DP solutions to the ERM problem:

$$\theta^* = \underset{\theta}{\mathrm{argmin}} \, \frac{1}{n} \sum_{i=1}^{n} - \log\left(1 + \exp\left\{-y_i \theta^\top x_i\right\}\right) := \underset{\theta}{\mathrm{argmin}} \, \frac{1}{n} \sum_{i=1}^{n} \ell_\theta(s_i, y_i)$$

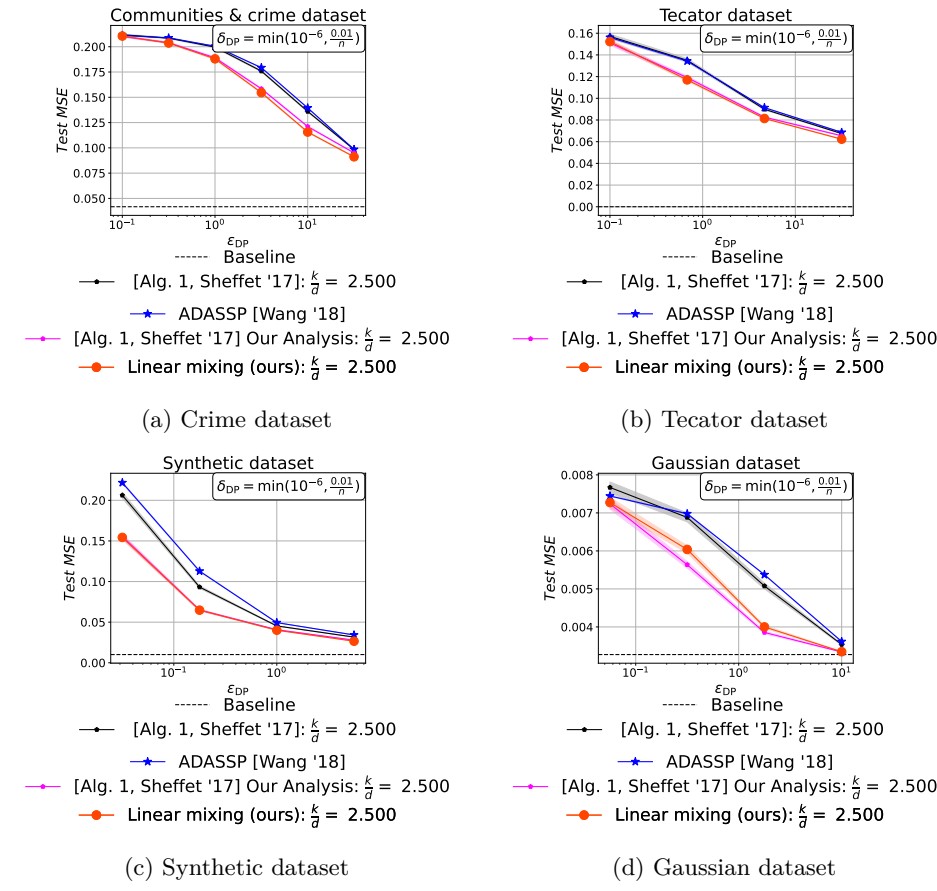

(a) Crime dataset

(b) Tecator dataset

(c) Synthetic dataset

(d) Gaussian dataset

Figure 2: Linear mixing performance on four linear regression tasks. The parameter $k$ was chosen to ensure negligible approximation error in the sketched solution.

and where $y_i \in \{-1, +1\}$ for all $i \in [n]$. Following [Huggins et al., 2017] (see also [Ferrando and Sheldon, 2025]), this problem can be solved by approximating $\ell_\theta(s, y)$ with a second-order polynomial $q(s) = b_0 + b_1 s + b_2 s^2$, i.e. $-\log(1 + \exp\{-s\}) \approx q(s)$, for $s \in \mathscr{I}$, where $\mathscr{I}$ satisfies $y\theta^\top x \in \mathscr{I}$ for all datapoints $(x, y)$, ensuring the surrogate is valid over the dataset. Substituting $s_i = y_i\,\theta^\top x_i$ and discarding the constant $b_0$, minimising the surrogate objective reduces to ordinary least squares with a response vector $\widetilde{Y} = -\frac{b_1}{2b_2}Y$, with $Y = (y_1, \ldots, y_n)^\top$. Therefore, we can invoke Algorithm 2, originally designed for linear regression, to obtain a $(\varepsilon, \delta)$-DP estimate for logistic regression. The utility of this solution follows via similar arguments to those of Theorem 2, with the replacement of $C_Y^2$ with $\left(\frac{b_1}{2b_2}\right)^2$ and with an extra factor due to the loss approximation and is presented in Appendix D. However, in this case, the complexity of sketching the data and then solving the linear system (for example, using a QR decomposition) will be $O(nkd + kd^2)$. In many cases, this one-shot approach is more computationally efficient than classical approaches that use iterative optimization techniques.

To demonstrate this point, we have tested our approach in a similar setting to that presented by Guo et al. [2020], where we (i) trained a CNN with DP stochastic gradient descent (DP-SGD) implemented using Opacus [Yousefpour et al., 2021] and (ii) used the pre-trained private embeddings for DP fine-tuning of a logistic head. The CNN architecture and training hyperparameters mirror those in [Guo et al., 2020] (see Appendix F for full details). We train the head using one of three methods: DP-SGD [Abadi et al., 2016, Yousefpour et al., 2021], objective perturbation as in [Guo et al., 2020], or with our approach (`Linear Mixing` as depicted in Algorithm 2). As shown in Figure 3, our approach delivers lower runtime than

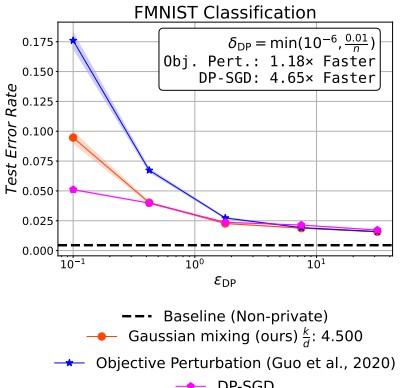
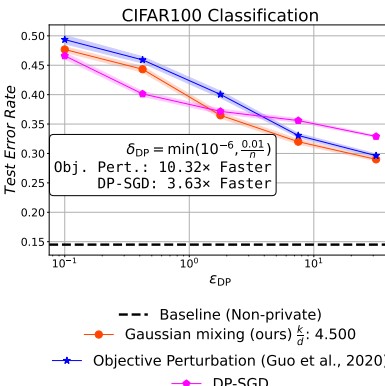

Figure 3: DP logistic regression using features from a privately trained CNN on binary subsets of Fashion-MNIST and CIFAR100. We fixed $k$ on $k = 4.5d$.

both baselines and achieves consistent accuracy gains over objective perturbation. It further exceeds DP-SGD accuracy for larger values of the privacy parameter $\varepsilon$. Full experimental details are provided in Appendix F.2.

## 6 Discussion and Conclusion

In this work, we revisited the Gaussian mixing mechanism originally introduced by Blocki et al. [2012] and later studied by Sheffet [2017, 2019]. We derived its RDP curve, which yields tighter bounds on the relationship between the noise parameter $\sigma$ and $(\varepsilon, \delta)$, thereby strengthening the privacy analysis of this mechanism. We further demonstrated the practical usefulness of this improved analysis by applying (GaussMix) to two distinct machine learning tasks and providing: (i) an improved algorithm for DP linear regression, (ii) an algorithm for DP logistic regression. The analysis used in the proof of Lemma 1 is tighter than that presented in [Sheffet, 2019]. Thus, it also offers performance improvement in other settings currently invoking the results of [Sheffet, 2019], such as [Bartan and Pilanci, 2023].

A key technical property that underpins the usefulness of (GaussMix) is its compatibility with formulations in which terms involving the random projection matrix S cancel in expectation, as described intuitively in (1). Identifying additional machine learning problems and formulations that naturally admit this structure and which facilitate an implementation using the Gaussian mixing mechanism is an interesting direction for future work, open the possibility for applying (GaussMix) beyond the currently studied linear regression settings. Moreover, this work focuses on the case where S is Gaussian. Extending the framework to other classes of projections, such as those studied by Woodruff [2014], Pilanci and Wainwright [2015, 2016], could enable computationally efficient implementation of (3).

**Broader Impact** The paper develops the DP guarantees for the Gaussian mixing mechanism. However, when used in practice, one needs to verify all the prior mathematical assumptions we made throughout in order to guarantee data privacy in practice.

## Acknowledgements

The authors thank the reviewers at NeurIPS 2025 for their valuable feedback on a preliminary version of this work. AS thanks Gautam Kamath for helpful discussions. The work of MS and KL was supported in part by ERC grant 101125913, Simons Foundation Collaboration 733792, Israel Science Foundation (ISF) grant 2861/20, Apple, and a grant from the Israeli Council of Higher Education. AW was supported in part by Simons Foundation Collaboration on Algorithmic Fairness. Views and opinions expressed are however those of the author(s) only and do not necessarily reflect those of the European Union or the European Research Council Executive Agency. Neither the European Union nor the granting authority can be held responsible for them.

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

# Appendix to the paper
# The Gaussian Mixing Mechanism:
# Rényi Differential Privacy via Gaussian Sketches

**Organization**  In Section A, we elaborate on notation used throughout this work. In Section B, we discuss a salient property of $\widetilde{\varepsilon}$ that appears in Algorithm 2. In Section C, we give proofs for the main theoretical results presented in the main text. Section D contains the utility guarantees for our differentially private logistic regression method in Section 5.2. We then discuss details of the various DP algorithms considered as baselines in Section E. Finally, we give details of the experiments in Section F, and additional numerical experiments in Section G.

## A  Notation

Given a matrix $A \in \mathbb{R}^{m \times n}$ we denote its elements by $A_{ij}$ and its column-stack representation by

$$\operatorname{vec}(A) \coloneqq (A_{11}, A_{21}, \ldots, A_{m1}, A_{12}, A_{22}, \ldots, A_{mn})^{\top}.$$

Random variables are denoted using sans-serif fonts (e.g., $\mathsf{X}, \mathsf{y}$), while their realizations are represented by regular italics (e.g., $X, y$). The $L_2$ norm of a vector $A \coloneqq (a_1, \ldots, a_d)$ is given by $\left( \sum_{i=1}^{d} a_i^2 \right)^{\frac{1}{2}}$ and is denoted by $\|A\|$. We denote the minimal and the maximal eigenvalues of a matrix $A$ by $\lambda_{\min}(A)$ and $\lambda_{\max}(A)$. We denote a PSD matrix $A$ by $A \succeq 0$ and a PD matrix by $A \succ 0$. We usually denote our dataset $\{x_i\}_{i=1}^{n}$ where each $x_i \in \mathbb{R}^d$ in the matrix form $X = (x_1, \ldots, x_n)^{\top}$. Then, we denote the $j$'th entry of $x_i$ by $x_i(j)$. The determinant of a matrix $A$ is denoted by $\det(A)$. The set of integer numbers from $1$ to $n$ is denoted by $[n]$. The all zeros column vector of size $d$ is denoted by $\vec{0}_d \coloneqq (0, \ldots, 0)^{\top}$. We denote by $\mathcal{N}(0, \mathbb{I}_{k_1 \times k_2})$ a $k_1 \times k_2$ matrix comprised of i.i.d. Gaussian elements with zero mean and unit variance. We denote by $\mathcal{N}_{\mathrm{sym}}(0, \mathbb{I}_d)$ a $d \times d$ symmetric matrix whose elements on the upper triangular matrix are i.i.d. and distributed accoring to $\mathcal{N}(0, 1)$. The Kronecker product between matrices $A$ and $B$ is defined via

$$A \otimes B \coloneqq \begin{pmatrix} A_{11}B & \ldots & A_{1m}B \\ & \ddots & \\ A_{n1}B & \ldots & A_{nm}B \end{pmatrix}.$$

The $k \times k$ identity matrix is denoted by $\mathbb{I}_k$.

## B  A central property of $\widetilde{\varepsilon}(\eta, \gamma, k, \delta)$

**Proposition 3.** *The function $\widetilde{\varepsilon}$ in (2) is non-negative. For fixed values of $k$, $\delta$, $\widetilde{\varepsilon}(\eta, \gamma, k, \delta)$ is a decreasing function in $\eta$, and satisfies $\widetilde{\varepsilon}(\eta, \gamma, k, \delta) \leq h(\gamma)$ for a monotonically decreasing function $h$ where $h(\gamma) \to 0$ as $\gamma \to \infty$.*

*Proof.* We now show that $\widetilde{\varepsilon}(\eta, \gamma, k, \delta) \geq 0$ and further that one can make $\widetilde{\varepsilon}(\eta, \gamma, k, \delta)$ as small as any desirable $\varepsilon$ by increasing $\eta$ and $\gamma$. We first note that $\varphi(\alpha; k, \gamma)$ is an upper bound on $D_{\alpha}(\mathcal{M}(X) \| \mathcal{M}(X'))$. Thus, following the validity of the conversion from RDP to DP of Canonne et al. [2020], the second term in $\widetilde{\varepsilon}(\eta, \gamma, k, \delta)$ provides an upper bound on the privacy parameter $\varepsilon$, and thus is non-negative. Since the first term in $\widetilde{\varepsilon}(\eta, \gamma, k, \delta)$ is non-negative we get that the entire expression is non-negative.

To prove that $\widetilde{\varepsilon}(\eta, \gamma, k, \delta)$ can be made arbitrarily small, we use the result of Corollary 1 which tells us that $\varphi(\alpha; k, \gamma) \leq \frac{k\alpha}{2\gamma^2}$ for $1 < \alpha \leq \frac{2}{5}\gamma$ and provided that $\gamma > \frac{5}{2}$. However, we note that the minimum in $\widetilde{\varepsilon}(\eta, \gamma, k, \delta)$ is upper bounded by

$$\min_{1 < \alpha < 2\gamma/5} \left\{ \frac{k\alpha}{2\gamma^2} + \frac{\log(3/\delta)}{\alpha - 1} \right\} \leq \frac{k}{5\gamma} + \frac{\log(3/\delta)}{\frac{2}{5}\gamma - 1}$$

which is derived by substituting $\alpha = 2\gamma/5$. Thus, this minimum is monotonically decreasing in $\gamma$ and can be made arbitrarily small by increasing $\gamma$. The result then follows since the first term in (2) is monotonically decreasing in $\eta$, and holds further in the case where $\eta = \frac{\gamma}{\sqrt{k}}$ by picking a sufficiently large $\gamma$. $\qquad\square$

## C   Proofs

### C.1   Proof of Lemma 1

The proof relies on the next lemma, which establishes the $\alpha$-Rényi divergence between two multivariate Gaussian distributions.

**Lemma 2** (Gil et al. [2013]). *Let $\mathsf{x}_1 \sim \mathcal{N}(\mu_1, \Sigma_1)$ and $\mathsf{x}_2 \sim \mathcal{N}(\mu_2, \Sigma_2)$. Then,*

$$D_\alpha(\mathsf{x}_1 \| \mathsf{x}_2) = \frac{\alpha}{2}(\mu_1 - \mu_2)^\top (\Sigma_1 + \alpha(\Sigma_2 - \Sigma_1))^{-1} (\mu_1 - \mu_2) \tag{4}$$

$$- \frac{1}{2(\alpha - 1)} \log \left( \frac{\det (\Sigma_1 + \alpha(\Sigma_2 - \Sigma_1))}{(\det (\Sigma_1))^{1-\alpha} (\det (\Sigma_2))^\alpha} \right)$$

*for all $\alpha$ such that $\alpha\Sigma_1^{-1} + (1-\alpha)\Sigma_2^{-1} \succ 0$.*

*Proof of Lemma 1.* Instead of analyzing GaussMix, we will analyze the transformed mechanism

$$\widetilde{\mathcal{M}}(X) := (\mathcal{M}(X))^\top = X^\top \mathsf{S}^\top + \sigma\xi^\top,$$

which, in terms of privacy, is equivalent to $\mathcal{M}(X)$. We note that $\widetilde{\mathcal{M}}(X)$ is a matrix of Gaussian random variables, where its columns are i.i.d. and each column has a covariance of

$$\mathbb{E}\left[ (X^\top \mathsf{s}_i + \sigma\xi_i)(X^\top \mathsf{s}_i + \sigma\xi_i)^\top \right] = X^\top X + \sigma^2 \mathbb{I}_d = \sum_{i=1}^n x_i x_i^\top + \sigma^2 \mathbb{I}_d$$

where we have denoted $\mathsf{S}^\top = (\mathsf{s}_1, \ldots, \mathsf{s}_k)$ and $\xi^\top = (\xi_1, \ldots, \xi_k)$. Thus, we first note that

$$\mathrm{vec}(\widetilde{\mathcal{M}}(X)) \sim \mathcal{N}(0, \mathbb{I}_k \otimes (X^\top X + \sigma^2 \mathbb{I}_d)).$$

Let $X'$ be our neighbor dataset that is different from $X$ in a single row. Throughout we assume that $X'$ is equivalent to $X$ except for one row which is zeroed out (see Section 3). We then show that the proof also covers the inverse case where one row of $X$ is zeroed out.

Without loss of generality, assume that the differing row is the first row of $X$. Thus, we first note that

$$\mathbb{I}_k \otimes (X^\top X + \sigma^2 \mathbb{I}_d) - \mathbb{I}_k \otimes (X'^\top X' + \sigma^2 \mathbb{I}_d) = \mathbb{I}_k \otimes (x_1 x_1^\top).$$

Let $\Sigma_1 := \mathbb{I}_k \otimes (X^\top X + \sigma^2 \mathbb{I}_d)$ and $\Sigma_2 := \mathbb{I}_k \otimes (X'^\top X' + \sigma^2 \mathbb{I}_d)$. Now,

$$\Sigma_1 + \alpha(\Sigma_2 - \Sigma_1) = \mathbb{I}_k \otimes \left( -\alpha x_1 x_1^\top + X^\top X + \sigma^2 \mathbb{I}_d \right)$$

and since $\mathbb{E}\left[ \mathrm{vec}(\widetilde{\mathcal{M}}(X)) \right] = \mathbb{E}\left[ \mathrm{vec}(\widetilde{\mathcal{M}}(X')) \right] = 0$ by using the algebraic identity $\det (\mathbb{I}_k \otimes A) = (\det (A))^k$ and by using (4) we get

$$D_\alpha(\mathcal{M}(X) \| \mathcal{M}(X')) = -\frac{k}{2(\alpha-1)} \log \left( \frac{\det \left( -\alpha x_1 x_1^\top + X^\top X + \sigma^2 \mathbb{I}_d \right)}{(\det (X^\top X + \sigma^2 \mathbb{I}_d))^{1-\alpha}(\det \left( X^\top X + \sigma^2 \mathbb{I}_d - x_1 x_1^\top \right))^\alpha} \right) .$$

$$\tag{5}$$

For an invertible matrix $A$, the matrix determinant lemma states that

$$\det \left( A + uv^\top \right) = \det (A) (1 + v^\top A^{-1} u).$$

Since the matrix $X^\top X + \sigma^2 \mathbb{I}_d$ is invertible whenever $\sigma^2 > 0$, this further tells us that the denominator of (5) can be simplified to

$$
\left(\det\left(X^\top X + \sigma^2 \mathbb{I}_d\right)\right)^{1-\alpha} \left(\det\left(X^\top X + \sigma^2 \mathbb{I}_d - x_1 x_1^\top\right)\right)^\alpha
$$
$$
= \left(\det\left(X^\top X + \sigma^2 \mathbb{I}_d\right)\right)^{1-\alpha} \left(\det\left(X^\top X + \sigma^2 \mathbb{I}_d\right)\right)^\alpha \left(1 - x_1^\top (X^\top X + \sigma^2 \mathbb{I}_d)^{-1} x_1\right)^\alpha
$$
$$
= \det\left(X^\top X + \sigma^2 \mathbb{I}_d\right) \left(1 - x_1^\top (X^\top X + \sigma^2 \mathbb{I}_d)^{-1} x_1\right)^\alpha.
$$

Thus, we further have

$$
\frac{\det\left(-\alpha x_1 x_1^\top + X^\top X + \sigma^2 \mathbb{I}_d\right)}{(\det\left(X^\top X + \sigma^2 \mathbb{I}_d\right))^{1-\alpha}(\det\left(X^\top X + \sigma^2 \mathbb{I}_d - x_1 x_1^\top\right))^\alpha}
$$
$$
= \left(\frac{\det\left(-\alpha x_1 x_1^\top + X^\top X + \sigma^2 \mathbb{I}_d\right)}{\det\left(X^\top X + \sigma^2 \mathbb{I}_d\right)}\right) \cdot (1 - x_1^\top (X^\top X + \sigma^2 \mathbb{I}_d)^{-1} x_1)^{-\alpha}.
$$

Similarly, we can apply the same determinant identity to $\det\left(-\alpha x_1 x_1^\top + X^\top X + \sigma^2 \mathbb{I}_d\right)$ and get

$$
\frac{\det\left(-\alpha x_1 x_1^\top + X^\top X + \sigma^2 \mathbb{I}_d\right)}{\det\left(X^\top X + \sigma^2 \mathbb{I}_d\right)} = 1 - \alpha x_1^\top (X^\top X + \sigma^2 \mathbb{I}_d)^{-1} x_1.
$$

Next, we note that this yields the next simplified form for $D_\alpha(\mathcal{M}(X) \| \mathcal{M}(X'))$:

$$
D_\alpha(\mathcal{M}(X) \| \mathcal{M}(X'))
$$
$$
= -\frac{k}{2(\alpha - 1)} \log\left((1 - \alpha x_1^\top (X^\top X + \sigma^2 \mathbb{I}_d)^{-1} x_1)(1 - x_1^\top (X^\top X + \sigma^2 \mathbb{I}_d)^{-1} x_1)^{-\alpha}\right)
$$
$$
= \frac{k}{2(\alpha - 1)} \log\left(\frac{(1 - x_1^\top (X^\top X + \sigma^2 \mathbb{I}_d)^{-1} x_1)^\alpha}{1 - \alpha x_1^\top (X^\top X + \sigma^2 \mathbb{I}_d)^{-1} x_1}\right). \tag{6}
$$

We note that the function $\frac{(1-t)^\alpha}{1-\alpha t}$ is a monotonically non-decreasing function of $t$ in the range $0 \le t < \frac{1}{\alpha}$ for $\alpha > 1$. To see this, note that

$$
\frac{\partial}{\partial t} \log\left(\frac{(1-t)^\alpha}{1-\alpha t}\right) = \frac{\partial}{\partial t}\{\alpha \log(1-t) - \log(1-\alpha t)\} = -\frac{\alpha}{1-t} + \frac{\alpha}{1-\alpha t} = \frac{\alpha(\alpha-1)t}{(1-t)(1-\alpha t)}
$$

which is positive in the range $0 \le t < \frac{1}{\alpha}$ (recall that $\alpha > 1$). Thus, to further simplify (6), we will try to find an upper bound on $x_1^\top (X^\top X + \sigma^2 \mathbb{I}_d)^{-1} x_1$. To that end, note that for a general symmetric positive-definite matrix $A$ we have

$$
x^\top A^{-1} x \le \frac{\|x\|^2}{\lambda_{\min}(A)}
$$

where equality is achieved whenever $x$ is the eigenvector of $A$ that correspond to $\lambda_{\min}(A)$. Then, using this relation with regard to $X^\top X + \sigma^2 \mathbb{I}_d \succ 0$ and using the identity

$$
\lambda_{\min}(X^\top X + \sigma^2 \mathbb{I}_d) = \lambda_{\min}(X^\top X) + \sigma^2
$$

yields

$$
x_i^\top (X^\top X + \sigma^2 \mathbb{I}_d)^{-1} x_i \le \frac{\|x_i\|^2}{\lambda_{\min}(X^\top X) + \sigma^2}, \quad \text{for all } i = 1, \ldots, n.
$$

Since we know that $\max_{i \in [n]} \|x_i\|^2 \le C_X^2$ we further have

$$
x_i^\top (X^\top X + \sigma^2 \mathbb{I}_d)^{-1} x_i \le \frac{C_X^2}{\lambda_{\min}(X^\top X) + \sigma^2}.
$$

This further leads to the next final upper bound on the Rényi divergence:

$$D_\alpha(\mathcal{M}(X)\|\mathcal{M}(X'))$$

$$= \frac{k}{2(\alpha-1)} \log\left( \frac{(1 - x_1^\top(X^\top X + \sigma^2\mathbb{I}_d)^{-1}x_1)^\alpha}{1 - \alpha x_1^\top(X^\top X + \sigma^2\mathbb{I}_d)^{-1}x_1} \right)$$

$$\leq \frac{k}{2(\alpha-1)} \log\left( \frac{(1 - \frac{\|x_1\|^2}{\lambda_{\min}(X^\top X)+\sigma^2})^\alpha}{1 - \frac{\alpha\|x_1\|^2}{\lambda_{\min}(X^\top X)+\sigma^2}} \right) \tag{7a}$$

$$= \frac{k\alpha}{2(\alpha-1)} \log\left( 1 - \frac{\|x_1\|^2}{\lambda_{\min}(X^\top X) + \sigma^2} \right) - \frac{k}{2(\alpha-1)} \log\left( 1 - \frac{\alpha\|x_1\|^2}{\lambda_{\min}(X^\top X) + \sigma^2} \right),$$

where (7a) requires that $\alpha < \frac{\lambda_{\min}(X^\top X)+\sigma^2}{\|x_1\|^2}$. Then, since a similar analysis holds when we replace $x_1$ with a general point $x_i$, the worst case divergence between $X$ and an $X_i'$ that is changed by zeroing out one entry $x_i$ is

$$\sup_{i\in[n]} D_\alpha(\mathcal{M}(X)\|\mathcal{M}(X_i'))$$

$$\leq \sup_{i\in[n]} \left\{ \frac{k\alpha}{2(\alpha-1)} \log\left( 1 - \frac{\|x_i\|^2}{\lambda_{\min}(X^\top X) + \sigma^2} \right) - \frac{k}{2(\alpha-1)} \log\left( 1 - \frac{\alpha\|x_i\|^2}{\lambda_{\min}(X^\top X) + \sigma^2} \right) \right\}$$

$$\leq \frac{k\alpha}{2(\alpha-1)} \log\left( 1 - \frac{C_X^2}{\lambda_{\min}(X^\top X) + \sigma^2} \right) - \frac{k}{2(\alpha-1)} \log\left( 1 - \frac{\alpha C_X^2}{\lambda_{\min}(X^\top X) + \sigma^2} \right)$$

$$\leq \frac{k\alpha}{2(\alpha-1)} \log\left( 1 - \frac{C_X^2}{\overline{\lambda}_{\min} + \sigma^2} \right) - \frac{k}{2(\alpha-1)} \log\left( 1 - \frac{\alpha C_X^2}{\overline{\lambda}_{\min} + \sigma^2} \right), \tag{8a}$$

where (8a) is again by the monotonicity of $\frac{(1-t)^\alpha}{1-\alpha t}$ and since $\lambda_{\min}(X^\top X) \geq \overline{\lambda}_{\min}$, where $\alpha \leq \min_i \left\{ \frac{\overline{\lambda}_{\min}+\sigma^2}{\|x_i\|^2} \right\} = \frac{\overline{\lambda}_{\min}+\sigma^2}{C_X^2}$ and the bound holds whenever $\overline{\lambda}_{\min} + \sigma^2 > C_X^2$. Finally, note that since $\alpha - 1 > 0$ for all $\alpha > 1$ and since $\alpha \log\left(1 - \frac{C_X^2}{\overline{\lambda}_{\min}+\sigma^2}\right) - \log\left(1 - \frac{\alpha C_X^2}{\overline{\lambda}_{\min}+\sigma^2}\right) \geq 0$ for all $\alpha > 1$ (this follows since the function is 0 for $\alpha = 1$ and since its derivative is positive) this upper bound is non-negative and is a valid upper bound on this divergence.

For the case where one row of $X$ is zeroed out, we note that we have $X'^\top X' + \sigma^2\mathbb{I}_d = X^\top X + \sigma^2\mathbb{I}_d + x_i x_i^\top$. Then, (6) is replaced with

$$D_\alpha(\mathcal{M}(X)\|\mathcal{M}(X')) = \frac{k}{2(\alpha-1)} \log\left( \frac{(1 + x_1^\top(X^\top X + \sigma^2\mathbb{I}_d)^{-1}x_1)^\alpha}{1 + \alpha x_1^\top(X^\top X + \sigma^2\mathbb{I}_d)^{-1}x_1} \right).$$

Now, we define the function $f(t;\alpha) = \log\left( \frac{(1-t)^\alpha}{1-\alpha t} \right) - \log\left( \frac{(1+t)^\alpha}{1+\alpha t} \right)$. Then, note that $f(0;\alpha) = 0$ and further since $\alpha > 1$ and $\alpha t < 1$ then

$$\frac{\partial}{\partial t} f(t;\alpha) = 2\alpha \left( \frac{1}{1-(\alpha t)^2} - \frac{1}{1-t^2} \right) \geq 0$$

and thus $f(t;\alpha) \geq 0$ for all $t < \frac{1}{\alpha}$, and we get that the maximum between the two divergences is always given by the case where $X'$ contains a zero row. Thus, by finding the $\sigma^2$ that makes (8a) equal to $\varepsilon$ we guarantee that our mechanism is $(\alpha, \varepsilon)$-Rényi-DP .

It remains to validate that the condition $\alpha\Sigma_1^{-1} + (1-\alpha)\Sigma_2^{-1} \succ 0$ holds. However, since throughout we have $\Sigma_2 = \Sigma_1 - x_i x_i^\top$ with $\Sigma_1 = X^\top X + \sigma^2\mathbb{I}_d$, by using the formulas for the inverse of a rank-1 update we get

$$\alpha\Sigma_1^{-1} + (1-\alpha)\Sigma_2^{-1} = \Sigma_1^{-1/2} \left( \mathbb{I}_d + (1-\alpha)\cdot\frac{\Sigma_1^{-1/2} x_i x_i^\top \Sigma_1^{-1/2}}{1 - x_i^\top\Sigma_1^{-1}x_i} \right) \Sigma_1^{-1/2}.$$

We note that since $\Sigma_1 \succ 0$, for this term to be positive definite it suffices for the middle matrix to be positive definite. However, since this matrix is a rank-1 update of $\mathbb{I}_d$, its eigenvalues are 1's and an additional eigenvalue that is given by

$$1 + (1-\alpha) \cdot \frac{\left\| \Sigma_1^{-1/2} x_i \right\|^2}{1 - x_i^\top \Sigma_1^{-1} x_i} = 1 + (1-\alpha) \cdot \frac{x_i^\top \Sigma_1^{-1} x_i}{1 - x_i^\top \Sigma_1^{-1} x_i}.$$

We note that this term is positive whenever $\alpha \leq \frac{1}{x_i^\top \Sigma_1^{-1} x_i}$. However, since $x_i^\top \Sigma_1^{-1} x_i \leq \frac{\|x_i\|^2}{\lambda_{\min} + \sigma^2}$ this inequality is satisfied by the restrictions we have on the domain of $\alpha$.

$\square$

## C.2    Proof of Corollary 1

*Proof.* We start by defining the difference function

$$\Delta(k, \alpha, \gamma) = \frac{k\alpha}{2\gamma^2} - \frac{k\alpha}{2(\alpha-1)} \log\left(1 - \frac{1}{\gamma}\right) + \frac{k}{2(\alpha-1)} \log\left(1 - \frac{\alpha}{\gamma}\right).$$

Our goal is to find when $\Delta(k, \alpha, \gamma) \geq 0$ for $1 < \alpha < \gamma$. We note multiplying by the positive factor $2\gamma^2(\alpha-1)$ and cancelling the term $k > 0$ gives the equivalent condition

$$G(\alpha) := \alpha(\alpha-1) - \alpha\gamma^2 \log\left(1 - \frac{1}{\gamma}\right) + \gamma^2 \log\left(1 - \frac{\alpha}{\gamma}\right) \geq 0, \qquad 1 < \alpha < w,$$

where $w$ will be specified shortly. On $\alpha = 1$ we further get $G(1) = 0$. Moreover,

$$G'(\alpha) = 2\alpha - 1 - \gamma^2 \log\left(1 - \frac{1}{\gamma}\right) - \frac{\gamma^2}{\gamma - \alpha},$$

and multiplying by $(\gamma - \alpha) > 0$ (recall that $\alpha < \gamma$) shows $G'(\alpha)$ has the same sign as the quadratic

$$H(\alpha) := (\gamma - \alpha)G'(\alpha)$$

$$= (2\alpha - 1)(\gamma - \alpha) - \gamma^2(\gamma - \alpha) \log\left(1 - \frac{1}{\gamma}\right) - \gamma^2$$

$$= -2\alpha^2 + \left(1 + 2\gamma + \gamma^2 \log\left(1 - \frac{1}{\gamma}\right)\right)\alpha - \gamma\left(1 + \gamma + \gamma^2 \log\left(1 - \frac{1}{\gamma}\right)\right).$$

We define the discriminant to be

$$\Delta_H = \left(1 + 2\gamma + \gamma^2 \log\left(1 - \frac{1}{\gamma}\right)\right)^2 - 8\gamma\left(1 + \gamma + \gamma^2 \log\left(1 - \frac{1}{\gamma}\right)\right)$$

which is non-negative. Thus, $H$ has two real roots

$$\alpha_{\max / \min} = \frac{\gamma^2 \log(1 - {}^1\!/\gamma) + 2\gamma + 1 \pm \sqrt{\Delta_H}}{4},$$

and $H(\alpha) \geq 0$ for $\alpha \in [\alpha_{\min}, \alpha_{\max}]$ since the coefficient of the quadratic term $H(\alpha)$ is negative. However, note that $\alpha_{\min} < 1/2$ for all $\gamma > 1$ and moreover $\alpha_{\max} > 1$ for $\gamma > 5/2$ and $\alpha_{\max} < \gamma$ for all $\gamma > 5/2$. Thus, since the derivative is positive and since $G(1) = 0$, setting $w := \alpha_{\max}$ yield $G'(\alpha) \geq 0$ for every $1 < \alpha < w$ and thus the inequality $G(\alpha) \geq 0$ holds throughout that interval, whenever $\gamma > 5/2$. The proof is completed since $\alpha_{\max} > \frac{2\gamma}{5}$ for all $\gamma > 1$.

$\square$

## C.3    Proof of Theorem 1

We recall that the sensitivity of the minimum eigenvalue $\lambda_{\min}(X^\top X)$ is $C_X^2$ (see, for example Sheffet [2017], Wang [2018]). Then, by using the standard formula of the Gaussian mechanism [Dwork et al., 2014, Appendix A] we get that $\widetilde{\lambda}$ is $(\sqrt{2\log(3.75/\delta)}/\eta, \delta/3)$ release

of $\lambda_{\min}(X^\top X)$. Using Lemma 1 and Proposition 1, we note that whenever $\lambda_{\min}(X^\top X)+\widetilde{\eta}^2 \geq \gamma$ the release of the output in both cases satisfies $(\widetilde{\varepsilon}, \delta/3)$-DP where

$$\widetilde{\varepsilon} = \min_{1<\alpha<\gamma} \left\{ \varphi(\alpha; k, \gamma) + \frac{\log(3/\delta) + (\alpha-1)\log(1-1/\alpha) - \log(\alpha)}{\alpha-1} \right\}.$$

The first case (when $\gamma \leq \tau$) trivially satisfies this. However, for the second case (whenever $\gamma > \tau$), this is satisfied only if $\widetilde{\eta}^2 + \lambda_{\min}(X^\top X) \geq \gamma$, which by using the inequality

$$\widetilde{\eta}^2 + \lambda_{\min}(X^\top X) = \gamma - \lambda_{\min}(X^\top X) + \eta C_X^2 \tau - \eta C_X^2 z + \lambda_{\min}(X^\top X) = \gamma + \eta C_X^2 \tau - \eta C_X^2 z$$

corresponds to having $z \geq \tau$ (we note that the case $\widetilde{\lambda} = 0$ immediately satisfies $\lambda_{\min}(X^\top X)+ \widetilde{\eta}^2 \geq \gamma$ since then we have $\widetilde{\eta}^2 = \gamma$). Thus,

$$P\left(\widetilde{\eta}^2 + \lambda_{\min}(X^\top X) \leq \gamma\right) = P\left(z \geq \tau\right) \leq \exp\left\{-\frac{\tau^2}{2}\right\} \leq \frac{\delta}{3}.$$

Then, using simple composition [Dwork et al., 2014, Chapter. 3] and substituting $\tau \geq \sqrt{2\log(3/\delta)}$ yields the desired result.

## C.4  Proof of Theorem 2

*Proof.* We first establish the performance of a method that adds noise with a general level $\sigma$, namely,

$$\theta_{\mathrm{Lin}} := ((\mathsf{S}X + \sigma\xi_1)^\top(\mathsf{S}X + \sigma\xi_1))^{-1}(\mathsf{S}X + \sigma\xi_1)^\top(\mathsf{S}Y + \sigma\xi_2).$$

Then, we can rewrite $\theta_{\mathrm{Lin}}$ in the next form

$$\theta_{\mathrm{Lin}} = \underset{\theta}{\arg\min} \left\| (\mathsf{S}, \xi_1, \xi_2) \left( \begin{pmatrix} Y \\ \vec{0}_d \\ \sigma \end{pmatrix} - \begin{pmatrix} X \\ \sigma \mathbb{I}_d \\ \vec{0}_d^\top \end{pmatrix} \theta \right) \right\|^2.$$

Now, since $\mathrm{rank}((X^\top, \sigma\mathbb{I}_d, \vec{0}_d)^\top) = d$ and since $\sigma^2 \geq 0$, by Pilanci and Wainwright [2015, Corollary. 2], whenever $k > \frac{c_0 d}{\chi^2}$ w.p. at least $1 - c_1 \cdot \exp\left\{-c_2 k\chi^2\right\}$ we have

$$L_{X,Y}(\theta_{\mathrm{Lin}}) + \sigma^2 \|\theta_{\mathrm{Lin}}\|^2 \leq (1+\chi)^2 \left( \|Y - X\theta^*(\sigma^2)\|^2 + \sigma^2 \|\theta^*(\sigma^2)\|^2 + \sigma^2 \right).$$

We note that this further implies that

$$L_{X,Y}(\theta_{\mathrm{Lin}}) \leq (1+\chi)^2 \left( \|Y - X\theta^*(\sigma^2)\|^2 + \sigma^2(1 + \|\theta^*\|^2) \right).$$

Thus, we can write

$$L_{X,Y}(\theta_{\mathrm{Lin}}) - (1+\chi)^2 L_{X,Y}(\theta^*) \leq (1+\chi)^2 \left( \|Y - X\theta^*(\sigma^2)\|^2 - \|Y - X\theta^*\|^2 + \sigma^2(1 + \|\theta^*\|^2) \right)$$

$$= O\left((1+\chi)^2 \sigma^2 \left(1 + \|\theta^*\|^2\right)\right)$$

where the last inequality is by Wang [2018, App. B.2]. Now, we note that in both of the cases of the algorithm the magnitude of the added noise is at most $\gamma(C_X^2 + C_Y^2)$, where $\gamma$ is determined by the calculation done in step 1. Thus, since the bound is monotonically increasing in $\sigma^2$ we can further use the upper bound

$$L_{X,Y}(\theta_{\mathrm{Lin}}) - (1+\chi)^2 L_{X,Y}(\theta^*) = O\left((1+\chi)^2 \gamma(C_X^2 + C_Y^2)\left(1 + \|\theta^*\|^2\right)\right). \tag{9}$$

We further note that

$$\varepsilon(\sigma,\gamma,k,\delta) = \frac{\sqrt{2\log(3.75/\delta)}}{\sigma} + \min_{1<\alpha<\gamma} \left\{ \varphi(\alpha; k, \gamma) + \frac{\log(3/\delta)+(\alpha-1)\log(1-1/\alpha)-\log(\alpha)}{\alpha-1} \right\}$$

$$\leq \frac{\sqrt{2k\log(3.75/\delta)}}{\gamma} + \min_{1<\alpha<2\gamma/5} \left\{ \frac{k\alpha}{2\gamma^2} + \frac{\log(3/\delta)}{\alpha-1} \right\}$$

$$= \frac{3\sqrt{2k\log(3.75/\delta)}}{\gamma}.$$

Thus, equating this upper bound to $\varepsilon$ suggests further that $\gamma = O\left(\frac{\sqrt{k\log(1/\delta)}}{\varepsilon}\right)$ and using this bound in (9) leads to

$$L_{X,Y}(\theta_{\text{Lin}}) - (1+\chi)^2 L_{X,Y}(\theta^*) = O\left((1+\chi)^2 \cdot \frac{\sqrt{k\log(1/\delta)}(C_X^2 + C_Y^2)}{\varepsilon} \cdot \left(1 + \|\theta^*\|^2\right)\right).$$

The proof is finished since this holds for any $\chi$ under the constraints in the theorem. $\qquad\square$

## D   Utility Guarantees for Logistic Regression

We now demonstrate utility guarantees on our method for DP logistic regression, presented in Section 5.2. Those derived similarly to Theorem 2, by considering both sources of errors: the error of approximating the objective with a polynomial and the empirical error of the linear regression solution. Throughout the proof, we denote by $\widehat{\theta}$ the private solution obtained by scaling the output of Algorithm 2 by $-\frac{b_1}{2b_2}$. We also let $\widetilde{\theta}^*$ denote the minimizer of the approximated loss, given explicitly by $-\frac{b_1}{2b_2}(X^\top X)^{-1}X^\top Y$. We further denote the empirical logistic loss via

$$L_{X,Y}(\theta) := -\frac{1}{n}\sum_{i=1}^{n}\log\left(1 + \exp\left\{-y_i x_i^\top \theta\right\}\right)$$

and the approximated empirical logistic loss via

$$\begin{aligned}
\widetilde{L}_{X,Y}(\theta) &:= b_0 + b_1\theta^\top\left(\frac{1}{n}X^\top Y\right) + b_2\theta^\top\left(\frac{1}{n}X^\top X\right)\theta \\
&= b_0 - \frac{b_1^2}{4nb_2}\|Y\|^2 + \frac{b_2}{n}\left\|-\frac{b_1}{2b_2}Y - X\theta\right\|^2 \\
&= b_0 - \frac{b_1^2}{4nb_2}\|Y\|^2 + \frac{b_2}{n}F\left(X, -\frac{b_1}{2b_2}Y, \theta\right).
\end{aligned}$$

We note that Corollary 2 guarantees that our logistic regression solution obtained by minimizing this surrogate is private. We now present the utility guarantees on this solution.

**Corollary 3.** *Assume that* $\|x_i\|_2^2 \leq C_X^2$, $|y_i|^2 \leq C_Y^2$ *and* $\left|y_i x_i^\top \widetilde{\theta}^*\right| \leq Q$ *and* $\left|y_i x_i^\top \widehat{\theta}\right| \leq Q$ *for all* $i \in [n]$ *and for some finite* $Q > 0$. *Let* $(b_0, b_1, b_2)$ *chosen such that*

$$\max_{s \in [-Q,Q]} \left|-\log(1 + e^{-s}) - (b_0 + b_1 s + b_2 s^2)\right| \leq q. \tag{10}$$

*Then, there exist universal constants* $c_0, c_1, c_2$ *such that for any* $\chi$ *satisfying* $k\chi^2 > c_0 d$ *the following holds with probability at least* $1 - c_1 \cdot \exp\left\{-c_2 k\chi^2\right\}$:

$$\begin{aligned}
&L_{X,Y}(\theta^*) - (1+\chi)^2 L_{X,Y}\left(\widehat{\theta}\right) - (1 + (1+\chi)^2)q + (1 - (1+\chi)^2)\left(b_0 - \frac{b_1^2}{4b_2}\right) \\
&\qquad = O\left((1+\chi)^2 \frac{\sqrt{k\log(1/\delta)}C_X^2}{n\varepsilon}\left(1 + \left\|\widetilde{\theta}^*\right\|^2\right)\right).
\end{aligned}$$

*Proof.* We first note that

$$L_{X,Y}(\theta^*) - (1+\chi)^2 L_{X,Y}\left(\widehat{\theta}\right) \le L_{X,Y}(\widetilde{\theta}^*) - (1+\chi)^2 L_{X,Y}\left(\widehat{\theta}\right) \tag{11a}$$

$$= L_{X,Y}(\widetilde{\theta}^*) - \widetilde{L}_{X,Y}(\widetilde{\theta}^*) + \widetilde{L}_{X,Y}(\widetilde{\theta}^*) - (1+\chi)^2 \widetilde{L}_{X,Y}\left(\widehat{\theta}\right)$$

$$+ (1+\chi)^2 \widetilde{L}_{X,Y}\left(\widehat{\theta}\right) - (1+\chi)^2 L_{X,Y}\left(\widehat{\theta}\right)$$

$$\le (1+(1+\chi)^2)q + \widetilde{L}_{X,Y}(\widetilde{\theta}^*) - (1+\chi)^2 \widetilde{L}_{X,Y}\left(\widehat{\theta}\right) \tag{11b}$$

$$= (1+(1+\chi)^2)q + (1-(1+\chi)^2)\left(b_0 - \frac{b_1^2}{4nb_2}\|Y\|^2\right) \tag{11c}$$

$$+ \frac{b_2}{n}\left(F\left(X, -\frac{b_1}{2b_2}Y, \widetilde{\theta}^*\right) - (1+\chi)^2 F\left(X, -\frac{b_1}{2b_2}Y, \widehat{\theta}\right)\right)$$

where (11a) is by the optimality of $\theta^*$, (11b) is by (10) and (11c) is by the definition of $\widetilde{L}_{X,Y}(\theta)$ and by the assumptions $\left|y_i x_i^\top \widetilde{\theta}^*\right| \le Q$ and $\left|y_i x_i^\top \widehat{\theta}\right| \le Q$. Then, the final result follows by using Theorem 2 and since in this case $|y_i| = 1$, thus $\|Y\| = n$ and $C_Y = 1$. $\quad\square$

When we take $\chi \ll 1$, the bound acquires an extra $2q$ term in the excess risk, introduced by the polynomial approximation.

# E   Algorithms for DP regression

## E.1   Linear regression

### E.1.1   AdaSSP

---
**Algorithm 3** AdaSSP [Wang, 2018]

---
**Require:** Dataset $(X, Y)$; Privacy parameters $\varepsilon, \delta$; Bounds: $\max_{i \in [n]} \|x_i\|^2 \le C_X^2, \max_{i \in [n]} |y_i|^2 \le C_Y^2$.

1: Calculate the minimum eigenvalue $\lambda_{\min}(X^\top X)$.

2: Privately release $\widetilde{\lambda}_{\min} = \max\left\{\lambda_{\min} + \frac{\sqrt{\log(6/\delta)}C_X^2}{\varepsilon/3}z - \frac{\log(6/\delta)}{\varepsilon/3}C_X^2, 0\right\}$ where $z \sim \mathcal{N}(0,1)$.

3: Set $\lambda = \max\left\{0, \frac{\sqrt{d\log(6/\delta)\log(2d^2/\varrho)}C_X^2}{\varepsilon/3} - \widetilde{\lambda}_{\min}\right\}$.

4: Privately release $\widetilde{X^\top X} = X^\top X + \frac{\sqrt{\log(6/\delta)}C_X^2}{\varepsilon/3}\xi_1$ for $\xi_1 \sim \mathcal{N}_{\text{sym}}(0, \mathbb{I}_d)$.

5: Privately release $\widetilde{X^\top y} = X^\top y + \frac{\sqrt{\log(6/\delta)}C_X C_Y}{\varepsilon/3}\xi_2$ for $\xi_2 \sim \mathcal{N}(0, \mathbb{I}_d)$.

6: **return** $\widetilde{\theta} \leftarrow \left(\widetilde{X^\top X} + \lambda\mathbb{I}_d\right)^{-1}\widetilde{X^\top y}$

---

### E.1.2 Algorithm 1 from Sheffet [2017] and our proposed modification

---

**Algorithm 4** Sheffet's Algorithm (Original)

---

**Require:** Dataset $(X, Y)$; Privacy parameters $\varepsilon, \delta$; Bounds: $\max_{i \in [n]} \|x_i\|^2 \le \mathrm{C}_X^2$, $\max_{i \in [n]} |y_i|^2 \le$ $\mathrm{C}_Y^2$; Hyperparameter $k$.

1: Compute $\lambda_{\min} := \lambda_{\min}((X, Y)^\top (X, Y))$.
2: Set $\gamma \leftarrow \frac{4(\mathrm{C}_X^2 + \mathrm{C}_Y^2)}{\varepsilon} \left( \sqrt{2k \log\left(\frac{8}{\delta}\right)} + 2\log\left(\frac{8}{\delta}\right) \right)$.
3: Sample $\mathsf{S} \sim \mathcal{N}(0, \mathbb{I}_{k \times n})$.
4: **if** $\lambda_{\min} > \gamma + \mathsf{z} + \frac{4(\mathrm{C}_X^2 + \mathrm{C}_Y^2)\log(1/\delta)}{\varepsilon}$ for $\mathsf{z} \sim \mathrm{Lap}\left(\frac{4(\mathrm{C}_X^2 + \mathrm{C}_Y^2)}{\varepsilon}\right)$ **then**
5:      **return** $\widetilde{\theta} \leftarrow \left((\mathsf{S}X)^\top (\mathsf{S}X)\right)^{-1} (\mathsf{S}X)^\top (\mathsf{S}Y)$
6: **else**
7:      Sample noises $\xi_1 \sim \mathcal{N}(0, \mathbb{I}_{k \times d})$, $\xi_2 \sim \mathcal{N}(0, \mathbb{I}_k)$.
8:      **return** $\widetilde{\theta} \leftarrow \left((\mathsf{S}X + \gamma \xi_1)^\top (\mathsf{S}X + \gamma \xi_1)\right)^{-1} (\mathsf{S}X + \gamma \xi_1)^\top (\mathsf{S}Y + \gamma \xi_2)$

---

**Algorithm 5** Sheffet's Algorithm with Our Analysis

---

**Require:** Dataset $(X, Y)$; Privacy parameters $\varepsilon, \delta$; Bounds: $\max_{i \in [n]} \|x_i\|^2 \le \mathrm{C}_X^2$, $\max_{i \in [n]} |y_i|^2 \le$ $\mathrm{C}_Y^2$; Hyperparameter $k$.

1: Compute $\lambda_{\min} := \lambda_{\min}((X, Y)^\top (X, Y))$.
2: Set $\gamma$ s.t. $\min_{1 < \alpha < \gamma} \left\{ \varphi(\alpha; k, \gamma) + \log\left(1 - \frac{1}{\alpha}\right) - \frac{\log(\alpha\delta)}{\alpha - 1} \right\} \le \varepsilon/2$.
3: Sample $\mathsf{S} \sim \mathcal{N}(0, \mathbb{I}_{k \times n})$.
4: **if** $\lambda_{\min} > \gamma + \mathsf{z} + \frac{4(\mathrm{C}_X^2 + \mathrm{C}_Y^2)\log(1/\delta)}{\varepsilon}$ for $\mathsf{z} \sim \mathrm{Lap}\left(\frac{4(\mathrm{C}_X^2 + \mathrm{C}_Y^2)}{\varepsilon}\right)$ **then**
5:      **return** $\widetilde{\theta} \leftarrow \left((\mathsf{S}X)^\top (\mathsf{S}X)\right)^{-1} (\mathsf{S}X)^\top (\mathsf{S}Y)$
6: **else**
7:      Sample noises $\xi_1 \sim \mathcal{N}(0, \mathbb{I}_{k \times d})$, $\xi_2 \sim \mathcal{N}(0, \mathbb{I}_k)$.
8:      **return** $\widetilde{\theta} \leftarrow \left((\mathsf{S}X + \gamma \xi_1)^\top (\mathsf{S}X + \gamma \xi_1)\right)^{-1} (\mathsf{S}X + \gamma \xi_1)^\top (\mathsf{S}Y + \gamma \xi_2)$

---

## E.2 Logistic Regression

### E.2.1 Objective Pertubation

---

**Algorithm 6** Objective Perturbation [Kifer et al., 2012]

---

**Require:** Dataset $(X, Y)$; privacy parameters $\varepsilon$ and $\delta$; Bound $\|x_i\| \le \mathrm{C}_X$ for all $i \in [n]$;

1: Set $\sigma = \frac{\sqrt{4\varepsilon + 8\log(2/\delta)}}{\varepsilon} \mathrm{C}_X$ and $\Delta = \frac{\mathrm{C}_X^2}{2\varepsilon}$.
2: Sample $\mathsf{b} \sim \mathcal{N}(0, \sigma^2 \mathbb{I}_d)$
3: **return** $\widetilde{\theta} \leftarrow \operatorname*{argmin}_{\theta} \left\{ \sum_{i=1}^n -\frac{1}{n} \log\left(1 + \exp\left\{-y_i x_i^\top \theta\right\}\right) + \frac{\mathsf{b}^\top \theta}{n} + \frac{\Delta}{2n} \|\theta\|^2 \right\}$.

---

# F    Experimental Details

All the linear regression experiments were run on 12th Gen Intel(R) Core(TM) i7-1255U, and all the logistic experiments were run on an NVIDIA A100 GPU.

## F.1   Linear Regression

For the linear regression experiments, we used four datasets. The first two are real-world datasets: the Tecator dataset [Thodberg, 2015] and the Communities and Crime dataset [Red-

mond and Baveja, 2002]. We have used a random train-test split of 80%/20% for generating a train and a test set.

The other two are synthetic datasets where the responses were generated via the linear model $y_i = x_i^\top \theta_0 + \sigma \xi_i$, with $\theta_0$ sampled as a unit vector uniformly from the $(d-1)$-dimensional sphere, $\xi_i \sim \text{Unif}(-1, 1)$, and $\sigma = 0.1$.

In the first synthetic dataset (termed *Gaussian dataset*), the parameters were $n = 8192, d = 512$, and the covariates were sampled as $x_i \sim \mathcal{N}(0, QQ^\top)$, where $Q \in \mathbb{R}^{d \times q}$ is a random orthogonal matrix with $q = 4$, ensuring the data lies on a 4-dimensional subspace. The matrix $Q$ was generated via QR decomposition of a random matrix with i.i.d. standard Gaussian entries.

The second synthetic dataset (termed the *synthetic dataset*) was constructed as follows. First, we sampled latent covariates $\widetilde{x}_i \sim \mathcal{N}(0, \mathbb{I}_2)$. Then, we generated final covariates using a two-layer neural network:

$$x_i = \phi(W_2 \phi(W_1 \widetilde{x}_i + b_1) + b_2),$$

where $\phi(\cdot)$ is the element-wise sigmoid function, $W_1 \sim \mathcal{N}(0, \mathbb{I}_{100 \times 2})$, $W_2 \sim \mathcal{N}(0, \mathbb{I}_{d \times 100})$, $b_1 \sim \mathcal{N}(0, 10^{-6} \cdot \mathbb{I}_{100})$, and $b_2 \sim \mathcal{N}(0, 10^{-6} \cdot \mathbb{I}_d)$. In our experiments, we have fixed $d = 2^9$.

For both synthetic datasets, the train and test sets were generated independently, using the same fixed $\theta_0$ but with independent covariates and additive noise.

In all cases, we normalized the training data so that the maximum $\ell_2$-norm of any training sample was 1. The test data was scaled using the same normalization factor as the training data.

The baseline (non-private) estimator was computed as $\widehat{\theta} = (X^\top X + \lambda \mathbb{I}_d)^{-1} X^\top Y$ for $\lambda = 10^{-6}$, ensuring invertibility in all cases. We report the mean squared error (MSE) for both the train and the test set, computed as the squared error in predicting $y_i$ via $x_i^\top \widehat{\theta}$. All results are averaged over 250 independent trials, and we report both the empirical means and confidence intervals.

### F.1.1 Algorithms

Our algorithm was implemented as described in Algorithm 2. The AdaSSP algorithm was implemented based on [Wang, 2018, Alg. 2], following the procedure detailed in Appendix E.1.1. Our second baseline, from [Sheffet, 2017, Alg. 1], was implemented according to the description in Appendix E.1.2. This implementation matches that of [Sheffet, 2017, Alg. 1], except for an adjustment to account for a factor of 2 in the parameter $w$, which arises due to using the zero-out neighboring definition rather than the replacement definition. In the variant of this baseline that incorporates our improved privacy analysis, we replaced the original noise calibration with bounds derived from Lemma 1, translated to $(\varepsilon, \delta)$-DP using the conversion provided in Proposition 1 (see also Appendix E.1.2).

### F.2 Logistic Regression

In this set of experiments, we trained a logistic regression classifier for a binary classification task without applying any regularization. Our non-private baseline is the standard `LogisticRegression` solver from the `sklearn.linear_model` library. The private baselines are the objective perturbation method (described in Appendix E.2), where the minimization is carried out using `torch.optim.LBFGS` with a maximum of 500 iterations and a tolerance of $10^{-6}$, following the setup of [Guo et al., 2020], and DP-SGD [Abadi et al., 2016] as implemented in Opacus Yousefpour et al. [2021] with a batch size of 1024, 10 epochs, and a learning rate of 0.5. While DP-SGD may benefit from hyperparameter tuning, our method requires only one; to avoid spending additional privacy budget on tuning, we use a fixed, reasonable configuration. We also fixed the parameter $k$ on $4.5d$.

We conducted experiments on the Fashion-MNIST [Xiao et al., 2017] and the CIFAR100 [Krizhevsky and Hinton, 2009] datasets, using the implementations provided in `torchvision.datasets`. From each dataset, we selected only the samples corresponding to classes 3 and 8, and relabeled them as $-1, 1$ to fit the binary classification setting. We used

the standard PyTorch train/test splits and normalized the training data by the maximum $L_2$ norm across all training samples, ensuring that each training sample has a norm of at most 1. The same normalization factor was then applied to the test set. The train and test loaders were generated using `torch.utils.data.DataLoader` with shuffling enabled. In Appendix G.2 we present additional simulations with the CIFAR10 [Krizhevsky and Hinton, 2009] and the MNIST [LeCun and Cortes, 2010] datasets.

The network architecture used is a compact convolutional neural network for RGB image classification. It consists of two convolutional layers with ReLU activations and max pooling, reducing the input to a 64-channel feature map of size 8×8. The flattened features are passed through a fully connected layer with 128 hidden units and ReLU, followed by a final linear layer that outputs class logits. In both of the experiments, we have first trained this network end-to-end using the DP-SGD primitive implemented in Opacus [Yousefpour et al., 2021], where we have set the clipping parameter to 4.0, learning rate to 0.001, the number of epochs to 20, and the batch size to 500.

Performance metrics are averaged over 50 independent runs, and as before, we report test accuracy along with confidence intervals. Runtime comparisons show the ratio of execution times for the largest simulated $\varepsilon$.

# G  Additional Experiments

## G.1  Linear Regression

We have simulated additional four datasets: the Boston housing dataset [Harrison Jr and Rubinfeld, 1978] that contains 506 measurements of 13-dimensional features with the goal of predicting house prices in the Boston area, the Wine quality dataset [wine, 2009] which contains 1359 measurements of 11-dimensional features, with the goal of predicting wine quality, the Bike sharing dataset [bike, 2019] with the goal of predicting the count of rental bikes, and another artificial dataset that follows the same description as that of the Gaussian dataset but now with i.i.d. features where the distribution of each entry is Unif$([-1, 1])$. The additional results are presented in Figure 4.

## G.2  Logistic Regression

We have simulated two additional datasets: the CIFAR10 [Krizhevsky and Hinton, 2009] and the MNIST [LeCun and Cortes, 2010] datasets, using the same logistic regression setting. The additional results are presented in Figure 5. Both settings demonstrate the computational improvement of our method, as well as utility improvement for a range of the simulated values of $\varepsilon$.

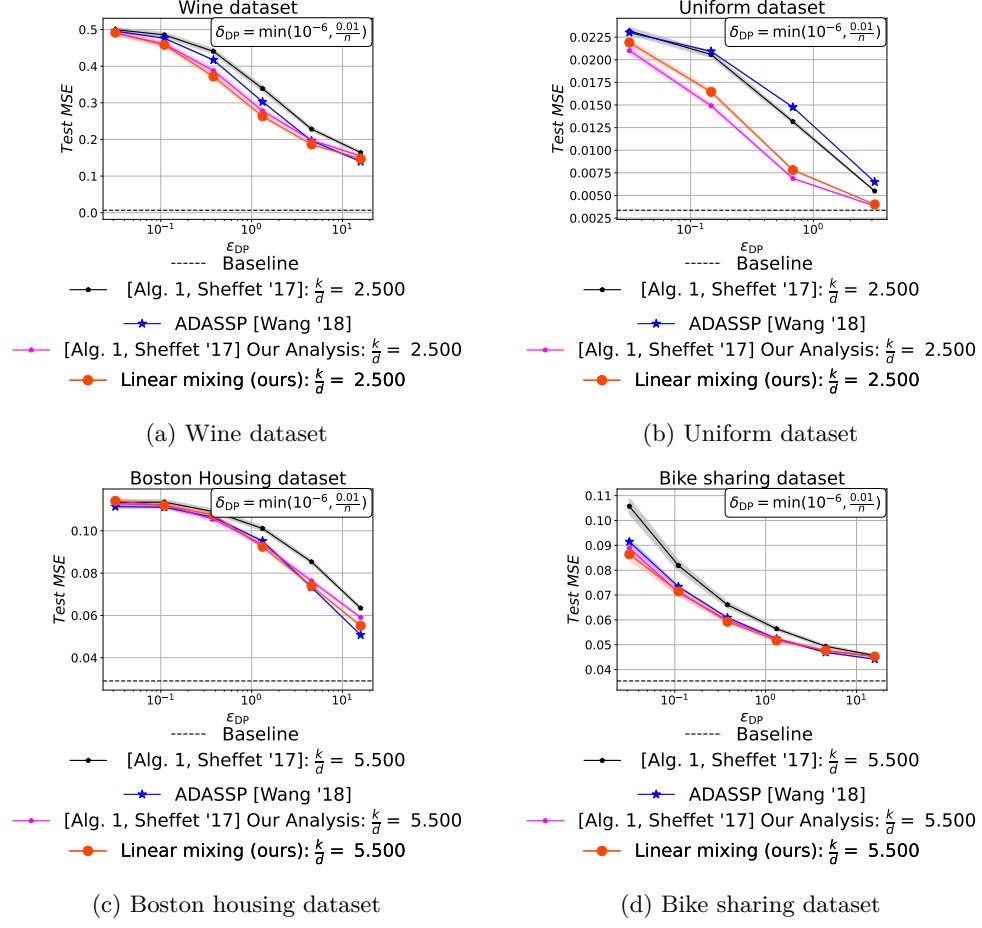

(a) Wine dataset        (b) Uniform dataset

(c) Boston housing dataset        (d) Bike sharing dataset

Figure 4: Linear mixing performance on the additional four linear regression tasks.

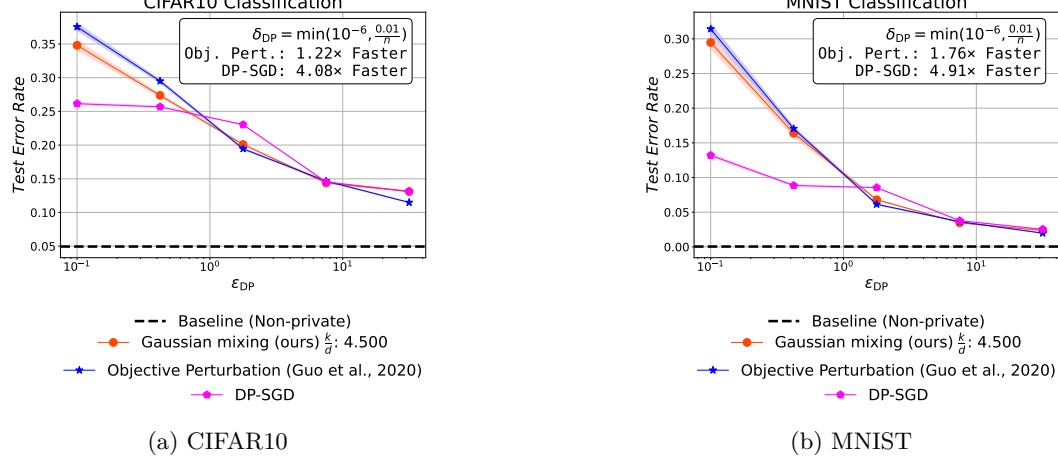

(a) CIFAR10        (b) MNIST

Figure 5: DP logistic regression using a privately trained CNN feature extractor on binary subsets of CIFAR10 and MNIST.

