# OpenReview forum: "The Gaussian Mixing Mechanism: Renyi Differential Privacy via Gaussian Sketches"
_NeurIPS.cc/2025/Conference — NeurIPS 2025 poster_

### Official Review · Reviewer_FmFk · 2025-06-19

**Clarity:** 4
**Significance:** 3
**Originality:** 2
**Rating:** 4
**Confidence:** 4

**Summary:**

This paper provides an improved privacy analysis for a mechanism that first applies a standard Gaussian random projection to the data matrix and then adds Gaussian noise. Using this sharper analysis, the authors obtain tighter utility bounds for differentially private linear and logistic regression, building on existing techniques. They also show that their method performs better in practice, demonstrating improved accuracy on standard linear regression datasets.

**Questions:**

The tightening of the RDP analysis still recovers the existing error bounds in the worst case. Other than a novel analysis, what is the main theoretical significance of the work that the authors would like to highlight?

**Ethical Concerns:**

["NO or VERY MINOR ethics concerns only"]

**Final Justification:**

The paper presents an interesting contribution by offering a novel privacy analysis tool. While this analytical approach is valuable, the paper itself does not introduce significantly new results in terms of utility; rather, it provides a refined method for analyzing privacy. Notably, similar techniques have appeared in implicit form in prior work, such as Blocki et al. (2012). The authors extend this line of work by proposing a more distribution-aware analysis that aligns with recent developments in moment-based relaxations of pure differential privacy, such as tCDP. Although this is a compelling and useful tool, it does not, in our view, merit a substantial increase in score, as it lacks major new contributions to utility bounds in the general case.

Blocki, J., Blum, A., Datta, A., & Sheffet, O. (2012, October). The johnson-lindenstrauss transform itself preserves differential privacy.

**Limitations:**

Yes

**Quality:**

4

**Strengths And Weaknesses:**

**Strengths:**

1. The core technique and accompanying theoretical analysis are both novel and thoughtfully developed. The novelty in the techniques pertaining to the analysis of a tighter RDP-DP conversion mechanism and deriving a tighter $\epsilon, \delta$-DP bound has been well established in this paper. These techniques would be interesting from the point of view of a nice conversion technique from RDP to DP bounds for non-gaussian like mechanisms. Such mechanisms are useful not only in linear regression but other random projection based mechanisms for which this paper provides a clear analysis.

2. The paper is well-organized and clearly communicates its main contributions. The exposition and proofs are accessible and precise. The proof-sketches within the paper is helpful along with a small ablation of different use-cases after each theorem.

3. Empirical results show the method performs favorably compared to existing approaches, suggesting practical promise. It can be seen that in practical datasets like Communities and Crime and the Teactor Dataset, the linear mixing performs better than all of the baselines the latest being . Moreover, in the synthetic and gaussian dataset versions, it is shown that the algorithm due to Sheffet 2017 [2] performs better but corresponding to the parameter calculations by the technique proposed by this paper, further underscoring the effectiveness of conversion technique in practice.

**Weaknesses:**

1. The paper could do a better job of explaining the theoretical significance of the method and how it compares to existing approaches. A clearer discussion of what sets it apart would be helpful. Most of the work seems to be derived from the analysis of Sheffet 2019 [1] paper and it is not clear how exactly are the authors differentiating in terms of their contribution. Other than proposing a more nuanced analysis of the random projection mechanism and further developments in further bounds, it is not clear what the overall contribution is as the utility bounds are mostly matching the previously derived bounds.

2. This connects to the previous point. It would also be useful to include some intuition or explanation for why the method performs better empirically—this could make the results more insightful and grounded. For instance, their improvements in parameters regimes does underscore well empirically, the authors should explicitly mention that in the experiments section and give further insights on how one can go about theoretically characterizing this improvement.

**Overall Assessment:**

This is a good paper with theoretical and empirical strengths. A deeper discussion of the method’s advantages and practical performance would make it even stronger and how it distinguishes itself from previous work.

[1] Or Sheffet. Old techniques in differentially private linear regression. In Algorithmic Learning Theory (ALT), pages 789–827. PMLR, 2019.
[2] Or Sheffet. Differentially private ordinary least squares. In Proceedings of the International Conference on Machine Learning (ICML), pages 3105–3114. PMLR, 2017.

---

> ### Author Rebuttal · Authors · 2025-07-31
>
> We thank the reviewer for their detailed and thoughtful feedback. We aim to address the points raised below.
>
> **Differentiation from [1, 2]:**
>
> - **Privacy Analysis via RDP:**
>
> While [1,2] analyze a variant of the Gaussian mixing mechanism using classical $(\varepsilon,\delta)$-DP techniques, our analysis leverages the RDP framework and leads to the following theoretical insights:
>
> 1. The RDP-based approach yields tighter bounds on the privacy parameters of the Gaussian mixing mechanism. These tighter guarantees directly translate into improved utility when applied to private linear regression.
>
> 2. We prove that the Gaussian mixing mechanism satisfies tCDP, a fact that was not known prior to our work. Moreover, and perhaps more surprisingly, this finding provides a novel example of a mechanism with a data-dependent covariance matrix that satisfies tCDP. Notably, such behavior is not exhibited by previously known mechanisms with data-dependent parameters that satisfy tCDP, such as the Gaussian smooth sensitivity.
>
> 3. For certain datasets $X$, our bounds are achieved with equality, giving a tight characterization of the privacy guarantees of the Gaussian mixing mechanism. As a direct consequence of our analysis, for input matrices $X$ such that $\underset{i\in [n]}{\sup} ||x_i||^2 \leq C_X^2$ and $\lambda\_{\min}(X^{\top}X) \geq C_X^2$, our bound is achieved with equality when $\frac{1}{C_X}X$ is semi-orthogonal. In particular, in this case, it holds that $\underset{i \in [n]}{\sup} \ x_i^\top (X^\top X + \sigma^2 \mathbf{I}_d)^{-1} x_i = \frac{C_X^2}{\sigma^2 + C_X^2}$ and all inequalities in our analysis are achieved with equality, resulting in tight privacy guarantees.
>
> - **Utility Guarantees for Gaussian Mixing in Private Linear Regression:**
>
> As a second theoretical contribution, we establish a formal utility guarantee for an algorithm that employs Gaussian mixing to perform private linear regression. This provides a meaningful extension beyond prior analyses in [1,2], which also connects to the utility theorem presented in [3].
>
>
> The current version of our work deliberately places greater emphasis on the refined privacy analysis, which is the central focus of this paper. An ongoing extension of this work aims to build upon our current utility analysis (Theorem 3) by providing a more detailed and rigorous utility comparison to complement our existing privacy results.
>
>  **Intuition for Empirical Improvements:**
>
> An intuition for these results can be formalized as follows:
>
> 1. Compared to the algorithms in [1, 2], our method provides tighter privacy guarantees and is therefore expected to yield strictly better performance in private linear regression tasks. To see this, note that for a fixed value of $\gamma$ (corresponding to a fixed noise level), the privacy parameter $\varepsilon$ achieved by our method is smaller whenever the condition$\frac{\sqrt{k}}{\gamma} < 2\sqrt{2\log(1/\delta)} + \frac{4\log(4/\delta)}{\sqrt{k}}$ is satisfied. As discussed in Section 4, the effective noise level in our scheme is given by $\frac{\gamma}{\sqrt{k}}$, which is typically much greater than one. This makes the above condition likely to hold in most practical settings. Consequently, for any fixed set of mechanism parameters, our method achieves a smaller $\varepsilon$—or equivalently, requires less noise to attain a given target $\varepsilon$—and is therefore expected to outperform the algorithm from [1, 2], as supported by our simulations.
>
> 2. Compared to the AdaSSP algorithm from [3], our current guarantees (presented in Theorem 3) align closely with those of AdaSSP in the regime where $\lambda_{\min}(X^{\top}X) \approx 0$. In this setting, both methods yield similar utility bounds. Notably, our method avoids the additional $\log(d^2/\varrho)$ dependence that appears in the AdaSSP analysis, and as such can provide improved guarantees—particularly when the parameter $\chi$ is small and $\varepsilon$ lies in a low to moderate privacy regime. This behavior is consistent with our empirical findings.
> In contrast, when $\lambda\_{\min}(X^{\top}X) \gg 0$, the analysis in [3] shows that AdaSSP enjoys a favorable inverse dependence on this eigenvalue, potentially leading to tighter bounds. Our method, on the other hand, relies on the minimum eigenvalue of the extended matrix $(X, Y)^{\top}(X, Y)$, which is always upper-bounded by $\lambda\_{\min}(X^{\top}X)$. Therefore, in the regime where $\lambda\_{\min}(X^{\top}X)$ is large while $\lambda\_{\min}((X, Y)^{\top}(X, Y))$ remains small, AdaSSP is expected to outperform our method.
> Lastly, there is a third regime, where both $\lambda\_{\min}(X^{\top}X) \gg 0$ and $\lambda\_{\min}((X,Y)^{\top}(X,Y)) \gg 0$. This situation arises when $Y$ is nearly orthogonal to the column space of $X$, such that the augmented matrix $(X, Y)$ is nearly full-rank. In this regime, the signal in $Y$ is poorly explained by $X$, and the underlying linear regression task becomes less meaningful and both methods perform poorly.
>
> **Main Theoretical Significance:**
>
> As highlighted earlier in our response, the core theoretical contribution of this work lies in the improved privacy analysis based on RDP and the conclusions it enables. This analysis yields not only tighter bounds on the privacy parameters of the Gaussian mixing mechanism but also a more nuanced characterization of its privacy behavior—both aspects that were not captured in prior work. Moreover, as our simulations show, this refined analysis is essential for allowing the sketching-based algorithm to compete effectively with AdaSSP.
>
>
> [1] Sheffet, Or. ``Differentially private ordinary least squares." International Conference on Machine Learning (ICML) PMLR, 2017.
>
> [2] Sheffet, Or. ``Old techniques in differentially private linear regression." Algorithmic Learning Theory (ALT) PMLR, 2019.
>
> [3]  Yu-Xiang Wang. Revisiting differentially private linear regression: Optimal and adaptive prediction \& estimation in unbounded domain. In Conference on Uncertainty in Artificial Intelligence (UAI), pages 93–103. AUAI Press, 2018.

---

> > ### Comment · Reviewer_FmFk · 2025-08-03
> >
> > Many thanks to the authors for addressing my comments! I appreciate the authors work and am happy with the current score.

---

### Official Review · Reviewer_UXxy · 2025-06-19

**Clarity:** 2
**Significance:** 3
**Originality:** 3
**Rating:** 4
**Confidence:** 4

**Summary:**

This paper studies the differential privacy mechanism of gaussian mixing mechanism, and improve the privacy guarantee by using techniques from renyi differential privacy. On a high level, the paper proposes a spectrum interpretation of the usefulness of the gaussian mixing mechanism, and provides a bound based on eigenvalues. As applications, the new analysis is applied on DP linear regression and DP logistic regression, and outperforms existing methods.

**Questions:**

- In figure 2, the pink line and red line are very close. Does this mean the improvement on the algorithm side is limited? I.e. algorithm in [1] is already good enough?
- It seems that the metric in differential private linear regression is the test MSE. In this case, have you considered to directly perturb the estimator? That is, you solve the OLS with the original data and perturb the final estimator. What is the advantage of gaussian mixing compared to this straight forward baseline?



[1] Sheffet, Or. "Differentially private ordinary least squares." International Conference on Machine Learning. PMLR, 2017.

**Ethical Concerns:**

["NO or VERY MINOR ethics concerns only"]

**Final Justification:**

I think this work provided good insights on using sketching techniques with differential privacy. The theoretical analysis is solid. I am leaning towards acceptance of this work.

**Limitations:**

Yes.

**Quality:**

3

**Strengths And Weaknesses:**

Pros:
- The study of Gaussian mixing mechanism is important. The high level idea of eigenvalues providing privacy is valuable. The theoretical part is well inspired and solid. The proof of spectrum argument seems promising to be applied in other stats/machine learning problems.
- The code is provided and the experiment results are great, outperforming the current SOTA results.

Cons:
- Theorems are not self-contained. For instance, in theorem 2, most of the contexts are missing and it is hard to parse. Also, algorithm 2 mentions assumption 1,2, but I could not find them in the paper.

---

> ### Author Rebuttal · Authors · 2025-07-31
>
> We thank the reviewer for highlighting the significance of our new analysis and for acknowledging the strength of our theoretical contributions.
>
> We also appreciate the comment regarding clarity and agree that some statements in the current version may be difficult to follow. In the revised version, we will revise the text to improve clarity and readability, as suggested.
>
> Regarding the questions raised:
>
> 1. As noted in the text, both the pink and red curves represent our newly developed analysis. The only substantive difference between the algorithm proposed in our work and the baseline in [1, 2] lies in how the smallest eigenvalue, $\lambda_{\min}((X, Y)^{\top}(X, Y))$, is estimated and used. In particular, when $\lambda_{\min}((X, Y)^{\top}(X, Y))$ is close to zero, the two versions behave similarly. Thus, the pink and red curves often overlap, with slight differences attributed to the different privacy parameter allocation throughout the algorithm. We emphasize that both variants stem from contributions introduced in our work, not from the results of [1].
>
> 2. The method referred to by the reviewer, known as output perturbation, can provide reasonably good private solutions under certain technical conditions (see, for example, [2, 4]). However, for private linear regression, the AdaSSP algorithm from [3] achieves the strongest guarantees on both empirical and test risk under minimal assumptions and is widely regarded as the leading baseline in this setting. For this reason, we selected AdaSSP as our baseline for the linear regression experiments.
>
>     For the logistic regression simulations, however, the output perturbation technique is generally considered less effective than the widely adopted objective perturbation approach (see, e.g., the discussions in [2] and [4]). For this reason, we selected objective perturbation as the primary baseline in our logistic regression experiments. To further strengthen our empirical comparison, we have included an additional DP-SGD baseline in this rebuttal (see the table below), which we also plan to incorporate into the final version of the manuscript.
>
>
> | Method               | ε = 0.1 | ε = 0.42 | ε = 1.78 | ε = 7.50 | ε = 31.62 | Slower by |
> |----------------------|---------|----------|----------|----------|------------|------------|
> | FMNIST (DP-SGD)      | 0.155   | 0.040    | 0.030    | 0.025    | 0.020      | 0.833      |
> | FMNIST (Obj. Pert.)  | 0.200   | 0.065    | 0.025    | 0.020    | 0.015      | 0.384      |
> | MNIST (Obj. Pert.)   | 0.350   | 0.155    | 0.070    | 0.040    | 0.025      | 0.666      |
> | MNIST (DP-SGD)       | 0.180   | 0.110    | 0.100    | 0.045    | 0.027      | 0.188      |
> | **FMNIST (Ours)**    | **0.125** | **0.030** | **0.025** | **0.014** | **0.0125** | --         |
> | **MNIST (Ours)**     | **0.315** | **0.130** | **0.080** | **0.045** | **0.028**  | --         |
>
>
> [1] Sheffet, Or. ``Differentially private ordinary least squares." International Conference on Machine Learning. PMLR, 2017.
>
> [2] Kifer, Daniel, Adam Smith, and Abhradeep Thakurta. ``Private convex empirical risk minimization and high-dimensional regression." Conference on Learning Theory. JMLR Workshop and Conference Proceedings, 2012.
>
> [3]  Yu-Xiang Wang. Revisiting differentially private linear regression: Optimal and adaptive prediction \& estimation in unbounded domain. In Conference on Uncertainty in Artificial Intelligence (UAI), pages 93–103. AUAI Press, 2018.
>
> [4] Chaudhuri, Kamalika, Claire Monteleoni, and Anand D. Sarwate. "Differentially private empirical risk minimization." Journal of Machine Learning Research 12.3 (2011).

---

> > ### Comment · Reviewer_UXxy · 2025-08-05
> >
> > Thanks for your response. My questions are addressed and I would like to maintain my positive evaluation.

---

### Official Review · Reviewer_HkTC · 2025-06-24

**Clarity:** 3
**Significance:** 2
**Originality:** 2
**Rating:** 3
**Confidence:** 4

**Summary:**

This paper proves privacy properties of what is called the Gaussian mixing mechanism -- "randomly" mixing the data values through multiplying by a Gaussian matrix followed by addition by another Gaussian matrix. By going to f-DP, tight privacy guarantees are provided. Then, it shows that this can lead to better algorithms for differentially private linear regression, and theoretical performance guarantees as well as empirical results are provided.

**Questions:**

none

**Ethical Concerns:**

["NO or VERY MINOR ethics concerns only"]

**Final Justification:**

I will retain my score "borderline reject". The paper has some interesting theory, but the empirical results are quite weak; consequently the contributions are perhaps not significant enough for NeuRIPS

**Quality:**

3

**Strengths And Weaknesses:**

Strengths:

+ The tight privacy analysis of the mixed Gaussian mechanism is well-done

Weaknesses:

- While the theoretical contributions are good, it was not completely clear to me what was mathematically novel -- the theory seemed to be quite standard applications of existing f-DP proof techniques.
- The empirical improvements are also somewhat marginal and quite small especially for small privacy budgets.

Overall I think the contributions are fairly marginal -- the theory is not terribly novel, and the empirical gains are very small.

---

> ### Author Rebuttal · Authors · 2025-07-31
>
> We thank the reviewer for their thoughtful comments and for recognizing the strength of our privacy analysis of the Gaussian mixing mechanism.
>
> Throughout the paper, we have aimed to highlight the main contributions, namely our refined RDP-based analysis of the Gaussian mixing mechanism and its implications for improving the performance of sketching-based algorithms for private linear regression. To that end, our primary objective was not to conduct an exhaustive empirical comparison against all existing methods, but rather to provide consistent evidence that our approach yields improvements over standard baselines in regimes where earlier analyses (e.g., [1, 2]) could not. While we acknowledge that the empirical gains may appear modest in some cases, we believe that consistently outperforming AdaSSP—particularly in settings where prior methods fail to do so—is a meaningful empirical contribution of our work. Moreover, as noted in our paper, the method proposed in [1, 2] remains effective in certain scenarios. Therefore, we believe that the fact that our approach achieves consistent and substantial performance improvements over this baseline across all datasets is another empirical contribution.
>
>
> **Comments Related to Theoretical Analysis:**
>
> We agree that our mathematical analysis by itself is not novel. However, we believe that our key conceptual and technical insight is that the privacy properties of this mechanism can be better understood, and its guarantees improved when viewed through the lens of RDP. This perspective enables us to derive tighter bounds than those previously known, and, furthermore, leads to two additional theoretical contributions:
>
>
>    1. Our bounds yield a closed-form characterization of the privacy parameters for certain inputs. To see this, as a direct consequence of our analysis, for input matrices $X$ such that $\underset{i\in [n]}{\text{sup}} ||x_i||^2 \leq C_X^2$ and $\lambda\_{\min}(X^{\top}X) \geq C_X^2$, our bound is achieved with equality when $\frac{1}{C_X}X$ is semi-orthogonal. In particular, in this case, it holds that $\underset{i\in [n]}{\text{sup}} \ x_i^\top (X^\top X + \sigma^2 \mathbf{I}_d)^{-1} x_i = \frac{C_X^2}{\sigma^2 + C_X^2}$ and all the inequalities in our analysis are achieved with equality, leading to tight privacy guarantees.
>
>    2. Our analysis establishes that the Gaussian mixing mechanism is tCDP. Importantly, we show that using the conversion between tCDP to $(\varepsilon, \delta)$-DP yields tighter privacy bounds than a direct $(\varepsilon, \delta)$-DP analysis—an outcome that is not immediate or obvious. In contrast, prior works such as [1, 2] rely on direct $(\varepsilon, \delta)$-DP arguments and derive weaker privacy guarantees. Furthermore, the fact that the Gaussian mixing mechanism satisfies tCDP enables one to leverage a broad body of existing theoretical results developed for this class of mechanisms. As a further insight, our work provides an example of a mechanism with a data-dependent covariance that satisfies tCDP. This stands in contrast to previously studied examples, which involve only data-dependent variances (see [4, 6]). We believe this observation is both nontrivial and potentially impactful, as it opens the door to further extend the previous developments from [4, 6] towards using richer data-dependent structures.
>
> In addition to the contributions previously discussed, we believe our results offer meaningful value to the differential privacy community—particularly given the widespread use of randomized sketches in the optimization literature. Our findings demonstrate that, in certain settings, sketching-based mechanisms can yield substantial improvements in both privacy-utility trade-offs and computational efficiency, outperforming several state-of-the-art baselines.
>
> **Comments Related to Significance of Empirical Results:**
>
> Our experimental results are designed to demonstrate the practical relevance of our improved privacy analysis, showing consistent utility gains across multiple linear regression tasks within the standard privacy regime of $\varepsilon \in [0.1, 10]$, a range widely accepted for $(\varepsilon, \delta)$-differential privacy. While we agree that the empirical gains might be small for certain cases, our focus was not the empirical study by itself but rather demonstrating that our method offers a mathematically rigorous refinement over the popular baselines introduced in [1, 2], and that it capable of achieving a non-trivial improvements over the widely used AdaSSP algorithm [5] in various settings. The ability of the Gaussian mixing mechanism to consistently outperform this strong baseline is, in our view, a meaningful contribution. Notably, this improvement holds uniformly across all datasets considered.
>
>
> We will ensure that these contributions are more clearly emphasized in the revised manuscript and thank the reviewer again for raising these points.
>
>
> [1] Sheffet, Or. ``Differentially private ordinary least squares." International Conference on Machine Learning (ICML) PMLR, 2017.
>
> [2] Sheffet, Or. ``Old techniques in differentially private linear regression." Algorithmic Learning Theory (ALT) PMLR, 2019.
>
> [3] Bun, Mark, et al. "Composable and versatile privacy via truncated CDP." Proceedings of the 50th Annual ACM SIGACT Symposium on Theory of Computing (SOTC). 2018.
>
> [4] Hendrikx, Hadrien, Paul Mangold, and Aurélien Bellet. "The relative Gaussian mechanism and its application to private gradient descent." International Conference on Artificial Intelligence and Statistics (AISTATS). PMLR, 2024.
>
> [5]  Yu-Xiang Wang. Revisiting differentially private linear regression: Optimal and adaptive prediction \& estimation in unbounded domain. In Conference on Uncertainty in Artificial Intelligence (UAI), pages 93–103. AUAI Press, 2018.

---

> > ### Comment · Reviewer_HkTC · 2025-08-04
> > **Thanks for response**
> >
> > Thank you for the response to my comments. I still think the actual gains are quite minimal (as shown by your experiments), and will maintain my score.

---

### Official Review · Reviewer_yhdg · 2025-06-26

**Clarity:** 3
**Significance:** 3
**Originality:** 3
**Rating:** 5
**Confidence:** 3

**Summary:**

This paper provides a tigher (Renyi DP) analysis for well-known Guassian mixing mechanism (GaussMix), which pre-multiplies the data matrix with a random Gaussian matrix (traditionally used for row mixing and reduction) before adding DP Gaussian noise.

Particularly, applications such as linear models which depend on cross-correlations of two matrices (e.g. features and labels) benefit from Gaussmix. GaussMix can be applied independently to each matrix, and the random matrix cancels out in expectation whp.

Their contribution is based on the observation that sketching itself can contribute to the privacy of outliers (in addition to dp noise) when the data matrix has large enough smallest Eigen value.

Their bound (Lemma 1, and Theorem 1) improves over Sheffet[2019]’s bound (which includes an additional term ($2 \log(4/\delta)/\gamma$)) and outperforms well-known AdaSSP baseline for linear regression. Additionally, it also boosts Sheffet[2017]’s algorithm.

GaussMix can also be used for logistic regression using a 2nd-order approximation of the loss function provided by [Huggings et al. 2017]. GaussMix with this new analysis provides improvements in the test errors over the objective perturbation baseline, in addition to significant computational saving (due to sketching).

**Questions:**

Comments/Clarifications:

1)	Did you observe any improvements in the experiments when data dependent noise is added with $\lambda_{min}$ >0?

2)	How does the performance of logistic regression with our bound compare to that of a single layer trained using DP-SGD?

3)	For logistic regression, any specific rational for privately obtaining (with DP-SGD) the embeddings needed for private finetuning? You could have privately finetune on the embeddings obtained non-privately?

**Ethical Concerns:**

["NO or VERY MINOR ethics concerns only"]

**Final Justification:**

My doubts have been resolved, authors should include experiments on DP-SGD with the next version.
Overall, a good contribution, and justifies acceptance.

**Limitations:**

Yes

**Quality:**

3

**Strengths And Weaknesses:**

Strength
1)	There is no prior work on the RDP analysis of GaussMix. The improvement in $\epsilon$ is tangible (Figure 1), and translates into improved accuracy in the regression experiments.
2)    The problem is a high impact. The GaussMix mechanism provides computational benefits in addition to privacy.
3)  The bound can be used independently  to also improve a prior algorithm ([Sheffet 2017]).
4)  It seems there are no numeric restrictions on the bound from Theorem 1. $\epsilon$'s can be made arbitrarily small, making the bound practical.
5)  Authors provide utility statement for linear regression which asymptotically match a baseline's utility statement.
5)	Generally well written. Authors make an effort to explain the  intuition --- DP noise boosts the minimum Eigen value of the sketched matrix to improve the privacy guarantee.

 Weakness:
1)	It's unclear how this method will stand against the gold standard baseline of DP-SGD when training a single layer logistic regression. The experiments do not include DP-SGD.  I think comparison with DP-SGD will only  strengthen the work, even if the method performs slightly inferior to DP-SGD in the worst case, if the computational benefits due to sketching are significant.

---

> ### Author Rebuttal · Authors · 2025-07-31
>
> We appreciate the reviewer’s positive feedback and acknowledgment of our work’s strengths. Please find below our detailed responses to the points you raised.
>
> We agree that DP-SGD is an important and widely used baseline in the literature, particularly due to its strong theoretical guarantees.
> To address your suggestion, we have extended our experiments to include DP-SGD as an additional baseline on two representative datasets. We selected a reasonable fixed set of DP-SGD hyperparameters to ensure a fair comparison without incurring additional privacy cost from hyperparameter tuning (which we view as a potential drawback of DP-SGD, especially compared to our proposed mechanism, which requires tuning only a single hyperparameter). As shown in the attached table, our method outperforms DP-SGD in terms of runtime, and in these specific settings, it also achieves higher test accuracy for some settings of $\epsilon$. We believe these results highlight the practical value of our method, particularly in scenarios where computational efficiency and simplicity are important considerations. We will add these additional simulations to the final version of our manuscript.
>
> Regarding the other points raised by the reviewer:
>
> 1. Usage with $\lambda_{\min} > 0$: Our method can be applied in cases where $\lambda_{\min} > 0$, and does not necessarily require that this eigenvalue is zero. However, note that our approach relies on $\lambda_{\min}((X,Y)^{\top}(X,Y))$ rather than $\lambda_{\min}(X^{\top}X)$. The former tends to be small in settings where $\underset{\theta}{\min} ||Y - X\theta||^2$ is small. This distinction implies that our method is expected to outperform AdaSSP when $\lambda_{\min}(X^\top X)$ is small, whereas in regimes where this eigenvalue is large, the guarantees in [2] suggest that AdaSSP benefits more due to its inverse dependence on $\lambda_{\min}(X^\top X)$. To clarify this point and the setting simulated in our work, we have added a table below reporting the observed minimum eigenvalues across all simulation settings.
>
> 2. Tuning Private Embeddings: We appreciate your observation that our method can also be applied to fine-tune embeddings obtained non-privately. In this work, we followed the setup proposed in [1], which motivated the need for privately fine-tuning a logistic head. Our goal was to demonstrate the effectiveness of our approach in this setting, demonstrating its effectiveness in a commonly used setup.
>
>
> | Method               | ε = 0.1 | ε = 0.42 | ε = 1.78 | ε = 7.50 | ε = 31.62 | Slower by |
> |----------------------|---------|----------|----------|----------|------------|------------|
> | FMNIST (DP-SGD)      | 0.155   | 0.040    | 0.030    | 0.025    | 0.020      | 0.833      |
> | FMNIST (Obj. Pert.)  | 0.200   | 0.065    | 0.025    | 0.020    | 0.015      | 0.384      |
> | MNIST (Obj. Pert.)   | 0.350   | 0.155    | 0.070    | 0.040    | 0.025      | 0.666      |
> | MNIST (DP-SGD)       | 0.180   | 0.110    | 0.100    | 0.045    | 0.027      | 0.188      |
> | **FMNIST (Ours)**    | **0.125** | **0.030** | **0.025** | **0.014** | **0.0125** | --         |
> | **MNIST (Ours)**     | **0.315** | **0.130** | **0.080** | **0.045** | **0.028**  | --         |
>
>
> | Dataset                           | Crime         | Tecator        | Synthetic      | Gaussian       | Wine           | Uniform        | Housing        | Bike           |
> |----------------------------------|---------------|----------------|----------------|----------------|----------------|----------------|----------------|----------------|
> | $$\lambda_{\min}(X^{\top}X)$$      | $4.3 \cdot 10^{-4}$  | $7.3 \cdot 10^{-16}$  | $1.5 \cdot 10^{-14}$  | $6.5 \cdot 10^{-16}$  | $7.6 \cdot 10^{-6}$  | $1.1 \cdot 10^{-13}$  | $2.4 \cdot 10^{-6}$  | $7 \cdot 10^{-17}$  |
> | $\lambda_{\min}((X,Y)^{\top}(X,Y))$ | $4.3 \cdot 10^{-4}$  | $6 \cdot 10^{-16}$    | $1.27 \cdot 10^{-14}$ | $2.4 \cdot 10^{-16}$  | $7.4 \cdot 10^{-6}$  | $9.6 \cdot 10^{-14}$  | $2.4 \cdot 10^{-6}$  | $3.5 \cdot 10^{-17}$ |
>
>
>
>
> [1] Chuan Guo, Tom Goldstein, Awni Hannun, and Laurens Van Der Maaten. Certified data removal from machine learning models. In Proceedings of the International Conference on Machine Learning (ICML). JMLR.org, 2020.362
>
> [2] Yu-Xiang Wang. Revisiting differentially private linear regression: Optimal and adaptive prediction \& estimation in unbounded domain. In Proc. 34th Conf. Uncertainty in Artif. Intell. (UAI), pages 93–404103. AUAI Press, 2018.

---

> > ### Comment · Reviewer_yhdg · 2025-08-04
> >
> > Thanks my concerns have been addressed. I will keep my positive score.  Good luck!

---

### Note · Authors · 2025-08-13

We thank the reviewers for their feedback on our work, and for discussions during the reviewer-author discussion period. We will incorporate the suggested revisions in the final version of this work. Here, we would like to take the opportunity to provide a general summary of our work as emphasized in most reviews.

- **Privacy analysis via R\'enyi-DP**:
    We provide an improved R\'enyi-DP analysis of the Gaussian mixing mechanism, leading to:
     1. **Tighter privacy bounds for the Gaussian mixing mechanism:** These guarantees translate into improved utility in private linear regression and can replace the analyses currently used in [1,2].
     2. **tCDP for Gaussian mixing:** We show that the mechanism satisfies tCDP, a fact that (to the best of our knowledge) was not previously established. We also find that the Gaussian mixing mechanism results in a distribution with a data-dependent covariance and this complements methods like Gaussian smooth sensitivity [4], which also has this property.
     3. **Tightness of our results:** We identify classes of input matrices $X$ for which our R\'enyi-DP bounds are tight, yielding a closed-form characterization of the mechanism’s privacy guarantees in these cases and revealing worst-case datasets for R\'enyi-DP.
- **Utility guarantees for private linear regression:** We establish a formal utility guarantee for an algorithm that employs Gaussian mixing in private linear regression, quantifying its performance under our improved privacy analysis.
-  **Empirical validation:** We demonstrate across multiple private linear regression settings that Gaussian mixing consistently improves performance relative to the widely used AdaSSP baseline [3]. Moreover, our refined analysis yields gains over the baselines introduced in [1,2], which—given their broad adoption—constitutes an independent contribution.

Taken together, these results extend [1,2] in a substantive way and demonstrate practical benefits in private linear regression.

[1] Sheffet, Or. Differentially private ordinary least squares, ICML 2017.

[2] Sheffet, Or. Old techniques in differentially private linear regression, ALT 2019.

[3] Yu-Xiang Wang. Revisiting differentially private linear regression: Optimal and adaptive prediction \& estimation in unbounded domain, Conference on Uncertainty in Artificial Intelligence (UAI) 2018.

[4] Bun, Mark, et al. Composable and versatile privacy via truncated CDP, Proc. 50th ACM STOC 2018.

---

### Decision · Program_Chairs · 2025-09-17

**Decision:**

Accept (poster)

**Comment:**

This paper studies the privacy properties of the Gaussian mixing mechanism, which multiplies the data with a random Gaussian matrix and then adds Gaussian noise. The authors user RDP analysis to provide tighter bounds for the privacy parameters of this mechanism, and show that it can be tight in certain settings. The paper also includes some experimental results showing some computational benefits while obtaining similar utility.

The reviewers are encouraged from the results in this paper, especially the tighter analysis. The authors should address the concerns of the reviewers' in their final version, for example, comparing to the standard DP-SGD baseline, and also highlighting the computational benefits, especially since the accuracy improvements are not big.

Given the above, I recommend acceptance.